# Cyclic stretch induces autophagy-mediated focal adhesion remodeling and activates mitochondria

Lukas Lövenich[1],*, Bahareh Rahimi[1],*, Henrique Baeta[1,2,3],*, Maithreyan Kuppusamy[1,2], Moritz Walkenbach[1], Frederik Rastfeld[1], Georg Dreissen[1], Ronald Wein[1], Jörg Höhfeld[4], Rudolf Merkel[1], Pitter F Huesgen[2,3,5], Bernd Hoffmann[1]

**Mammalian cells are continuously exposed to internally generated or externally applied mechanical stimuli. Mechanosensitive proteins enable cells to sense mechanical stress and induce protective mechanisms like autophagy and cytoskeletal reorientation. However, how these contribute to cellular and tissue adaptations remains largely unknown. Here, we studied the response of rat smooth muscle cells (A7r5) to uniaxial cyclic stretch. Stretching induced autophagy and adaptive actin fiber reorientation. Inhibiting autophagy using chloroquine or expressing a Bag3 (T285D-S289D) phosphosite mutant that impairs chaperone-assisted selective autophagy (CASA) delayed reorientation. Proteomic analysis revealed a depletion of cytoskeletal and focal adhesion proteins after stretching, which was attenuated by autophagy inhibition. Stretching caused a reduction in focal adhesion (FA) size, and the remodeled FAs reoriented perpendicularly to the strain direction. Concurrently, prolonged stretching activated mitochondria, and inhibiting mitochondrial ATP synthesis slowed actin reorientation, suggesting that mitochondrial activity supports the mechanoresponse. Our findings highlight the role of autophagy and mitochondria in the structural remodeling of cells upon adaptation to mechanical stress.**

## Introduction

Enduring mechanical stress is intrinsic to organism physiology, essential for adaptation to environmental conditions, and critical for survival (Dufort et al, 2011; Yusko & Asbury, 2014). Cells recognize mechanical cues through mechanosensitive proteins, also called mechanosensors, that can trigger intracellular signaling cascades to induce a mechanoresponse (Höhfeld et al, 2021). Mechanosensors are located at cell sites prominently exposed to tension, including focal adhesions (FAs), adherens junctions (AJs), or the cytoskeleton, and usually initiate signaling cascades through force-induced conformational changes involving multiple cycles of unfolding and refolding (Yao et al, 2016). Unfolding can alter protein–protein interactions by exposing cryptic domains (Paci & Karplus, 1999; Yao et al, 2016), by generating catch bonds (Huang et al, 2017), or by either opening (Hirata et al, 2014; Sadhanasatish et al, 2023) or disrupting (Ehrlicher et al, 2011) protein-binding sites. These conformational changes can also expose (Gordon et al, 2015) or protect (Santos et al, 2014) proteolytic sites, modify phosphorylation residues (Sawada et al, 2006), activate ion channel activity (Uchida et al, 2021), or stabilize protein conformations (Li & Springer, 2018), thereby translating mechanical stimuli into biochemical signals. However, excessive force can irreversibly destroy the structure of mechanosensors, rendering refolding impossible and triggering proteotoxic stress (Höhfeld et al, 2021). To counteract this, chaperone-assisted selective autophagy (CASA) acts as an essential force-protective mechanism, whereby irreversibly damaged filamins are degraded in smooth muscle cells (Ulbricht et al, 2013), striated muscle cells (Arndt et al, 2010), and human skeletal muscles (Ulbricht et al, 2015). Force-induced dephosphorylation of the cochaperone Bag3 triggers the formation of the CASA complex, which selectively targets denatured proteins to LC3B (Map1lc3b)-coated phagophores (Ottensmeyer et al, 2024). Bag3, along with Synpo2, SNARE membrane fusion, and VPS (vacuolar protein sorting) complexes (Ulbricht et al, 2013), facilitates the engulfment of damaged proteins into autophagosomes, which are then trafficked toward degradative lysosomes. Noticeably, Bag3 coordinates the degradation of filamins with their transcription (Ulbricht et al, 2013) and synthesis (Kathage et al, 2017) to sustain the cytoskeleton under mechanical stress. Despite

[1]Forschungszentrum Jülich, Institute of Biological Information Processing, IBI-2, Jülich, Germany    [2]Forschungszentrum Jülich, Central Institute for Engineering, Electronics and Analytics, ZEA-3, Jülich, Germany    [3]Institute for Biology II, University of Freiburg, Freiburg, Germany    [4]Institute for Cell Biology, University of Bonn, Bonn, Germany    [5]CIBSS-Centre for Biological Signaling Studies, University of Freiburg, Freiburg, Germany

Correspondence: b.hoffmann@fz-juelich.de; pitter.huesgen@biologie.uni-freiburg.de
Maithreyan Kuppusamy's present address is Department Metabolism, Senescence and Autophagy, Research Centre One Health, University Hospital Essen, Essen, Germany
*Lukas Lövenich, Bahareh Rahimi, and Henrique Baeta contributed equally to this work

these insights, the full range of CASA substrates remains elusive, emphasizing the need for their identification to fully understand its role in mechanical adaptation.

Cytoskeletal reorientation is a prominent adaptive mechanoresponse. In vivo, endothelial cells in blood vessels position their actin fibers parallel to the direction of blood flow (Wong et al, 1983). When subject to fluid shear stress in vitro, endothelial cells actively align their actin fibers in the flow direction (Sakamoto et al, 2010), but perpendicular to the strain direction upon uniaxial cyclic stretch (Lövenich et al, 2021). Upon stretching, mammalian cells reorient not only their actin fibers (Jungbauer et al, 2008; Faust et al, 2011; Das et al, 2022), but all cytoskeletal structures (actin, microtubules, and intermediate filaments) (Zielinski et al, 2018; Püllen et al, 2023) in a frequency (Lee et al, 2010)-, amplitude (Faust et al, 2011)-, and time-dependent manner (Lövenich et al, 2021). This adaptive mechanoresponse can be observed in subconfluent cells (Lövenich et al, 2021), cell monolayers (Noethel et al, 2018), and multilayered cell systems (Rübsam et al, 2023). Force-induced cytoskeletal dynamics are energy-demanding and rely on the hydrolysis of adenosine triphosphate (ATP), which poses a direct link between cellular mechanics and metabolism (Bays et al, 2017; DeWane et al, 2021). Furthermore, disruption of the actin cytoskeleton, mediated through the phosphoinositide 3-kinase (PI3K) pathway, boosts glycolysis by release of aldolase A from filamentous actin (F-actin) (Hu et al, 2016). Similarly, inducing stress fiber disassembly through a reduction of cell substrate stiffness leads to down-regulation of glycolysis by proteasomal degradation of phosphofructokinase (PFK) (Park et al, 2020). Moreover, propagating waves of the cell cortex driven by the Ras/PI3K/F-actin network are locally enriched with glycolytic enzymes providing ATP for protrusions and migration (Zhan et al, 2025). However, although cyclic stretching of cells can influence mitochondrial function (Kim et al, 2018; Han et al, 2023), it remains unclear whether stretching activates mitochondria or whether their ATP synthesis supports the mechanoresponse.

Cyclic stretching induces autophagy, and inhibiting this process slows the adaptive cytoskeletal reorientation of dispersed A7r5 rat smooth muscle cells (Lövenich et al, 2021). We hypothesized that cells must dispose of damaged mechanosensors via autophagy to be able to remodel their mechanosensitive structures and properly reorient under strain. In this study, we focused on actin fiber reorientation and autophagy induction in A7r5 cell monolayers, investigating how their proteome and mechanosensitive structures adapt after uniaxial cyclic stretching. We found that the strain-induced mechanoresponse involves autophagy-mediated focal adhesion remodeling and reorientation perpendicular to the strain direction, alongside the actin fibers. In addition, we observed that prolonged stretching activates mitochondria and that their ATP synthesis supports the strain-induced mechanoresponse.

# Results

## Monolayer mechanoresponse resembles subconfluent cell behavior

We investigated whether A7r5 rat smooth muscle cells grown as monolayer (ML) cells, with fully established cell–cell contacts, reorient their cytoskeleton in response to cyclic strain like dispersedly grown subconfluent cells (SC) (Lövenich et al, 2021), which only form cell–matrix contact sites. First, we examined the focal adhesions (FAs) and adherens junctions (AJs) of A7r5 cells grown to different densities on fibronectin-coated elastomers (Fig 1). As expected, immunostaining for paxillin (Pxn) revealed prominent FA structures on the edge of actin fibers in both SC and ML cells (Fig 1A). Immunostaining for α-catenin (Ctnna1), representing AJs, showed that AJs were present on the cell–cell contacts of ML but not detected in SC cells (Fig 1B). Co-immunostaining for the FA proteins paxillin (Pxn) and vinculin (Vcl) was performed to study FAs in ML and SC cells (Fig S1A). Analysis revealed that the area of Pxn spots was identical in ML and SC cells (Fig S1B), and the area of Vcl spots was also identical in ML and SC cells (Fig S1D). Colocalization analysis revealed that the fraction of FAs covered by Pxn was smaller in ML compared with SC cells, whereas no difference was observed in the fraction of FAs covered by Vcl between ML and SC cells (Fig S1C and E). The FA areas were also identical in ML and in SC cells (Fig S1F), indicating that ML and SC cells have comparable FA structures.

We then compared the response of A7r5 ML after uniaxial cyclic stretching with prior published reorientation in A7r5 SC cells (Lövenich et al, 2021). In unstretched SC and ML cells, the orientation of actin fibers was randomly distributed, with no preferred orientation (mean orientation: ML: 45°; SC: 46°) (Fig 1C and D). After stretching, ML cells reoriented their actin fibers perpendicular to the strain direction (indicated by arrowheads), within the same timeframe previously observed and published for SC cells (Lövenich et al, 2021) (Fig 1C and D). After 30 min of stretching, the actin fiber orientation in most cells was >45° to the strain direction and continuously progressed toward 90° after 1- and 4-h stretching, in both cell densities (Fig 1D). For visualization, upper and lower 95% confidence intervals (CI95) of both SC and ML cells were plotted for all time points (Fig S1H–K). Thus, A7r5 SC and ML cells exhibit an identical adaptive mechanoresponse to uniaxial cyclic stretching, marked by time-dependent actin fiber reorientation perpendicular to the stretch direction.

**Uniaxial stretch induces autophagy**

To determine whether cyclic strain induces autophagy in A7r5 ML, LC3B (Map1lc3b)—a protein targeted to autophagosome (AP) membranes upon autophagy induction (Kabeya et al, 2000)—was stained after uniaxial cyclic stretching (Fig 2A). LC3B spots per cell, which we interpret as APs, were compared with (+) or without treatment (untreated) with the AP turnover inhibitor chloroquine (CQ). As expected, CQ treatment led to an increase in AP numbers in unstretched control cells. However, stretching caused further AP accumulation, reaching a maximum of 170 spots/cell after 4 h of stretching (Fig 2A and B), revealing force-induced autophagy induction. The dynamics of autophagosome formation and turnover under stretch became apparent in cells not treated with CQ. Unstretched, untreated control cells exhibited a low number of APs (1.4 spots/cell, indicated by red arrowheads) (Fig 2A and B). After 10 min of

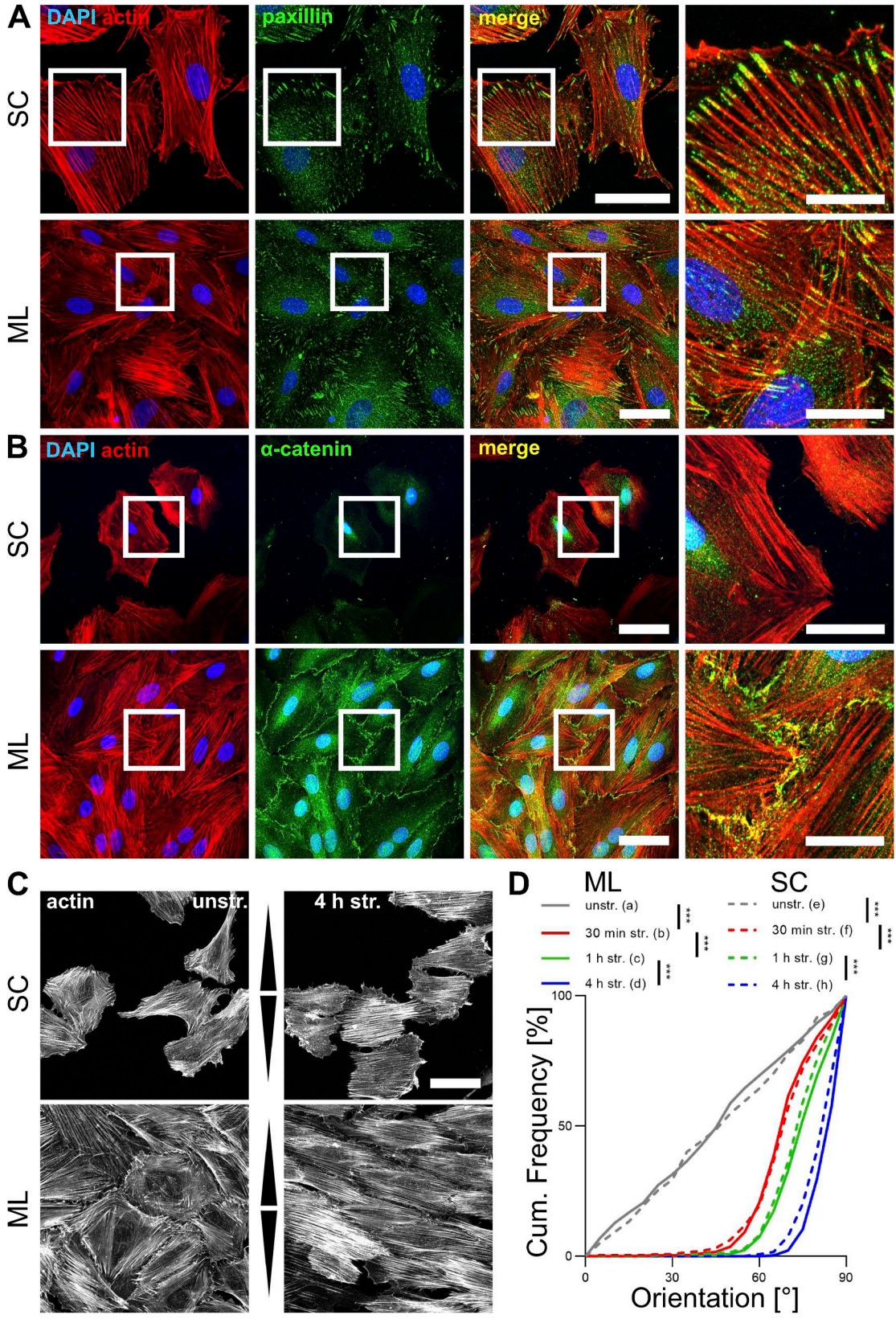

**Figure 1. Mechanosensitivity of A7r5 monolayer (ML) and subconfluent (SC) cells.**

A7r5 rat smooth muscle cells were grown on fibronectin-coated elastomeric PDMS substrates (unstretched). **(A, B)** Actin cytoskeleton (red) and (A) paxillin (Pxn) or (B) α-catenin (Ctnna1) immunostaining (green) as markers of focal adhesions or adherens junctions, respectively. The scale bar is 50 μm in overview and 20 μm in zoom

stretching, the number of APs peaked threefold. Prolonged stretching revealed a reverse pattern, with the number of APs decreasing below that of the unstretched control cells after 4 h of stretching (1.0 spots/cell). This result indicates that uniaxial cyclic stretching induces a transient burst of autophagy.

Next, we used immunoblotting to investigate the shift from cytosolic LC3B-I to its phosphatidylethanolamine-conjugated form LC3B-II, targeted to AP membranes upon the induction of autophagy (Mizushima & Yoshimori, 2007) in A7r5 ML after stretching (Fig 2C and D). Stretching led to a transient increase in the normalized ratio of LC3B-II relative to LC3B-I after 30 min of stretching (30-min str.) compared with the unstretched control (unstr.). The difference decreases with prolonged stretching after 1 and 4 h but remains significantly increased compared with the unstretched control (Fig 2C and D). After 10 min of stretching, LC3B-I and LC3B-II both were slightly increased, whereas LC3B shifted more to LC3B-II by parallel CQ treatment (Fig S2A and B). After 4 h of stretching without CQ, the protein levels of both LC3B-I and LC3B-II were decreased (Fig S2C and D), indicating a productive turnover of APs, in agreement with the low number of APs observed by imaging of the LC3B immunostainings (Fig 2A and B). In contrast, after 4 h of stretching with CQ treatment, LC3B-I was depleted, whereas LC3B-II accumulated (Figs 2C and D and S2C and D), consistent with the accumulation of APs observed by imaging (Fig 2A and B). These results provide additional evidence that uniaxial cyclic stretching transiently induces autophagy in A7r5 ML, characterized by a rapid formation of APs within 10–30 min of stretching followed by their turnover at later time points.

### Autophagy maintains the mechanoresponse to uniaxial stretch

To investigate the relationship between strain-induced transient autophagy induction and adaptive reorientation, we analyzed the orientation of actin fibers in A7r5 ML after uniaxial cyclic stretching, with or without CQ treatment (Fig 3). CQ treatment had no effect on the random orientation of actin fibers in unstretched control cells (Fig 3A and B). Upon stretching, CQ treatment slowed down the actin fiber reorientation at each time point, compared with untreated stretched cells (Fig 3A–C). For visualization, upper and lower 95% confidence intervals (CI95) of untreated and CQ-treated (+CQ) ML for all time points were plotted (Fig S2E–H).

The cochaperone Bag3 was previously identified as a critical regulator of the strain-induced reorientation in SC cells (Lövenich et al, 2021). Recently, we observed that Bag3 dephosphorylation triggered by mechanical stress, including cyclic stretch, represents an essential phospho-switch for chaperone-assisted selective

autophagy (CASA) (Ottensmeyer et al, 2024). We examined the role of CASA in the actin fiber orientation upon uniaxial cyclic stretching in A7r5 SC cells transiently expressing human Bag3 mutant carrying T285 and S289 substitutions to aspartic acid (T285D-289D) compared with previously published Bag3-WT actin fiber reorientation from Lövenich et al (2021) added here for direct comparison. The mutant mimics the phosphorylated form of human Bag3 that is unable to execute CASA (Ottensmeyer et al, 2024). Cells were cotransfected with GFP plasmids as transfection control, and only the orientation of GFP-expressing cells was analyzed (Fig 4). The expression of Bag3 (T285D-S289D) or Bag3-WT (Lövenich et al, 2021) had no effect on the equally distributed actin fibers in unstretched control cells (Fig 4A and B). Upon stretching, cells expressing the inactive Bag3 (T285D-289D) mutant showed delayed reorientation at each time point, compared with those expressing Bag3-WT (Fig 4A–C). For visualization, upper and lower 95% confidence intervals (CI95) of Bag3-WT and Bag3 (T285D-S289D) for all time points were plotted (Fig S2I–L). Taken together, these results indicate that autophagy and, more specifically, Bag3-dependent CASA contribute to the actin fiber reorientation in A7r5 cells after uniaxial cyclic stretching.

### Uniaxial stretch induces proteome adaptation

Stretching promotes AP formation and autophagy-mediated actin fiber reorientation in A7r5 ML cells. To evaluate whether the strain-induced autophagy leads to the degradation of damaged mechanosensors, we analyzed the proteome of A7r5 ML cells by quantitative mass spectrometry after 4 h of uniaxial cyclic stretching, with or without CQ treatment (Fig 5). A principal component analysis (PCA) based on 4,064 proteins quantified in all samples clearly separated stretched and unstretched control cells in the first component, which explains the largest portion of variation in the dataset (36.7%). The second component, explaining 12% of the variance, separated replicates from CQ-treated and untreated cells (Fig 5A). Comparison between stretched and unstretched control cells revealed significant changes in protein abundance (LIMMA moderated $t$ test, adjusted $P$ <0.05) (Fig 5B). The 1,004 proteins depleted after stretching included the mechanosensors Pxn, Vcl, actinins (Actn1 and Actn4), talin (Tln1), zyxin (Zyx), Vasp, p130Cas (Bcar1), Flii, Lamc2, and filamins (Flna, Flnb, and Flnc), among others (Table S1). Functional GO-term enrichment analysis indicated that the proteins depleted after stretching were mostly associated with cytoskeletal, microtubule, supramolecular fiber, and focal adhesion categories (Fig 5C). This suggested a more general, rather than protein-specific, effect of stretching on cytoskeletal and adhesion structures, particularly on FAs (Figs 5B and

---

images. **(C)** Actin fiber staining (white) in A7r5 monolayer (ML) compared with prior published subconfluent (SC) cells (Lövenich et al, 2021) after 4 h of uniaxial cyclic stretching (4 h str.) or in unstretched control cells (unstr.). Arrowheads illustrate stretch direction. The scale bar is 50 $\mu m$. **(D)** Actin fiber orientation is represented as an angular distribution from 0° to 90° in the direction of stretch (0° means in stretch direction; 90° means perpendicular to stretch direction), plotted as cumulative frequencies. Results for SC cells were extracted from Lövenich et al (2021) and are reused here for direct comparison. For ML cells, n represents the number of images taken with typically >15 cells each; $n_a$ = 61, $n_b$ = 60, $n_c$ = 60, and $n_d$ = 46; for SC cells, n represents the number of analyzed cells in each image; $n_e$ = 292, $n_f$ = 254, $n_g$ = 259, $n_h$ = 244 reused from Lövenich et al (2021). The Kolmogorov–Smirnov test was performed to test differences between conditions (***$P$ ≤ 0.001). Source data are available for this figure.

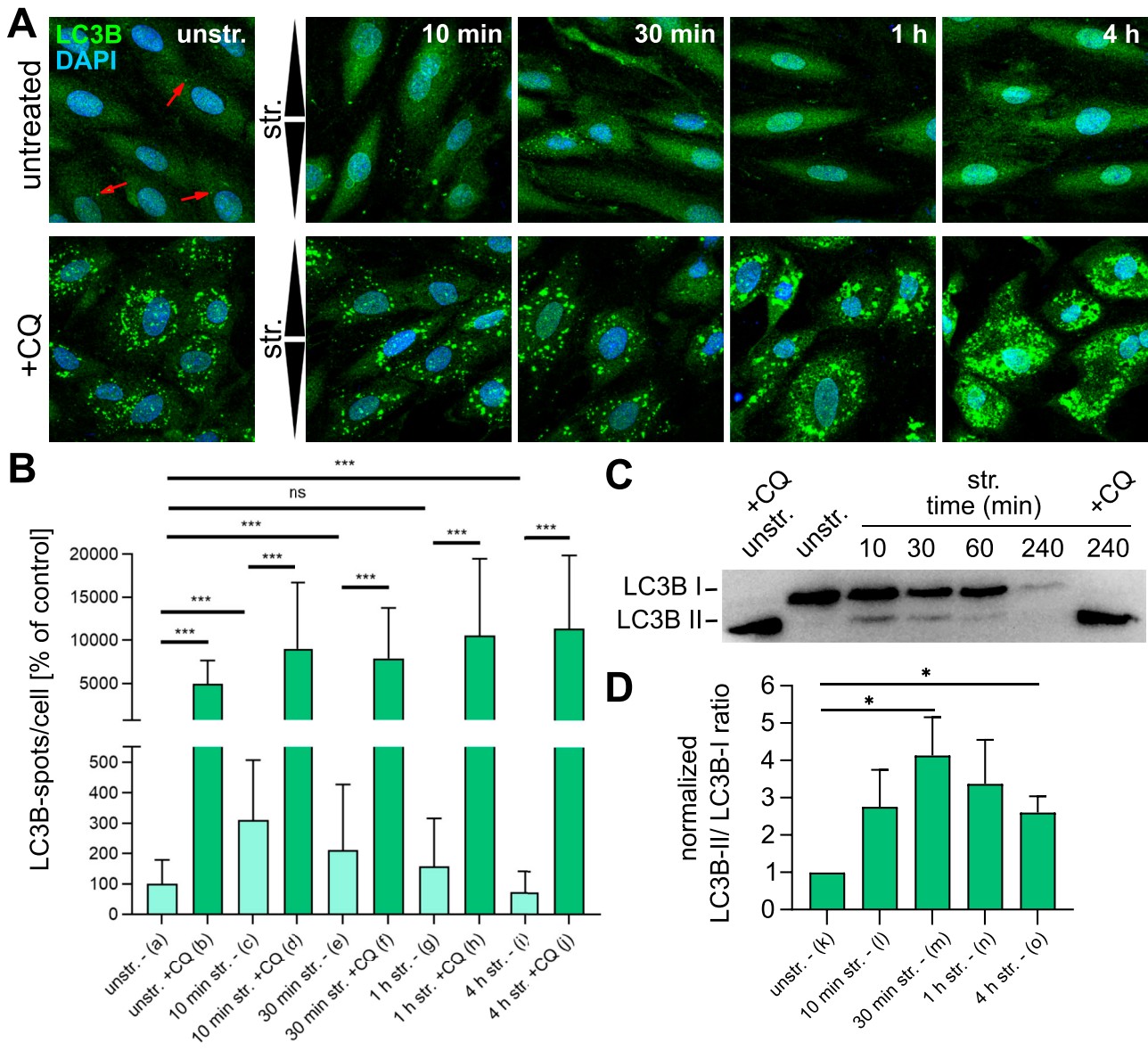

**Figure 2. Autophagy induction in A7r5 monolayers upon uniaxial cyclic stretching.**
**(A)** Immunostaining of LC3B (green) in A7r5 cell monolayers (ML) after uniaxial cyclic stretching (str.) and in unstretched controls (unstr.), with (+) or without chloroquine (CQ) treatment. Arrowheads illustrate stretch direction. The scale bar is 50 $\mu$m. **(B)** Quantification of LC3B spots per cell after stretching, normalized to the unstretched untreated control cells (unstr. –). Data are represented as the mean ± SD (n represents the number of analyzed cells; $n_a$ = 216, $n_b$ = 148, $n_c$ = 135, $n_d$ = 165, $n_e$ = 148, $n_f$ = 143, $n_g$ = 133, $n_h$ = 150, $n_i$ = 141, and $n_j$ = 142). The Mann–Whitney test was performed to test differences between conditions (ns $P > 0.05$, ***$P \leq 0.001$). **(C, D)** Quantification of the LC3B-II to LC3B-I protein level ratio by Western blot analysis in A7r5 ML lysates after 10, 30 min, and 1 and 4 h of stretching (str.) and in unstretched control cells (unstr.) with (+CQ) or without CQ treatment, as well as 4 h of stretching +CQ treatment (4 h str. +CQ). The ratios of the band intensities of LC3B-II to LC3B-I were plotted as relative to the unstretched untreated control (unstr. -) as mean value ± SEM ($n_{k-o}$ = 6 independent biological repeats). Statistical test was carried out using a two-tailed unpaired $t$ test with Welch's correction (*$P \leq 0.05$).
Source data are available for this figure.

C and S3). We therefore investigated whether the depletion of FA proteins, many of which are mechanosensors, after stretching was mediated by autophagy. Among the proteins that were depleted after stretching without CQ (str. versus unstr., adj. $P < 0.05$), 69 proteins were associated with the GO-term FA (GO:0005925) (Fig S3). 25 of these accumulated upon stretching in the presence of CQ (str.+CQ versus str., $P < 0.05$), indicating that their strain-induced degradation was attenuated by autophagy inhibition (Fig 5E). In

addition, CASA regulators Bag3, Hspa8, Stub1, and Map1lc3b were also depleted after stretching (Fig 5B), consistent with the involvement of Bag3-mediated CASA in the mechanoresponse of A7r5 SC (Fig 4). Among the 1,024 proteins that accumulated after stretching, 285 proteins were associated with mitochondria (GO: 0005739), the most enriched category (Fig 5D). Other proteins that accumulated after stretching included several integrins (Itga1, Itga3, Itga7, Itgav, Itgb1, and Itgb5), keratins (Krt14, Krt16, Krt42, and

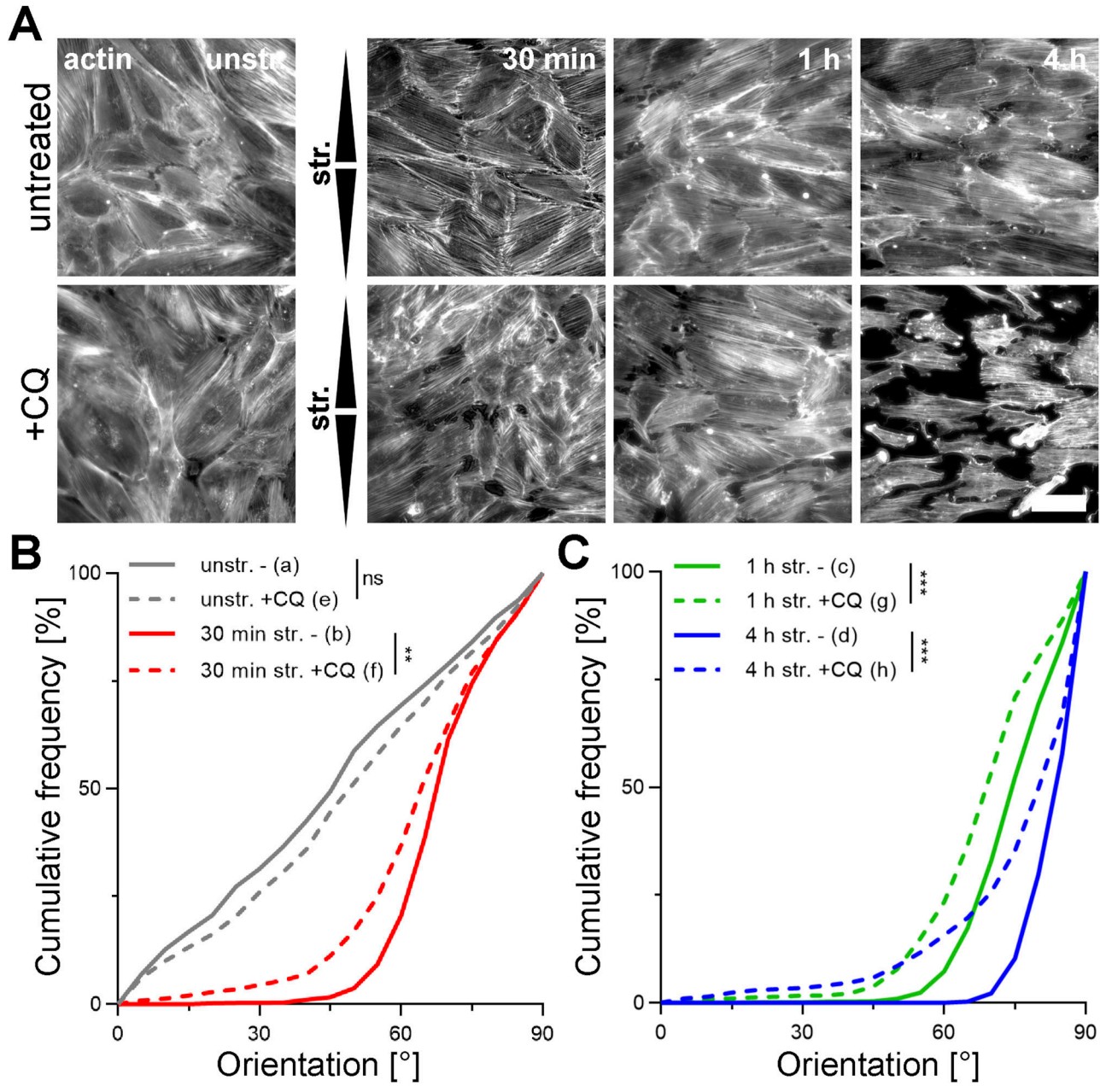

**Figure 3. Mechanoresponse of A7r5 monolayers is mediated by autophagy.**
**(A)** Staining for actin fibers (white) in A7r5 monolayers (ML) after 30 min, and 1 and 4 h of uniaxial cyclic stretching (str.) and in unstretched control cells (unstr.), without (−) or with chloroquine (+CQ) treatment. Arrowheads illustrate stretch direction. The scale bar is 50 $\mu m$. **(B, C)** Actin fiber orientation is represented as angular distribution from 0° to 90° in the stretch direction (0° means in stretch direction; 90° means perpendicular to stretch direction), plotted as cumulative frequencies observed in A7r5 ML after 30 min, and 1 and 4 h of stretching and in unstretched control cells (unstr.), with (+CQ) or without (−) CQ treatment. (n represents the number of images taken with typically >15 cells each, $n_{a-d}$; see Fig 1 ($n_{a-d}$), $n_e$ = 46, $n_f$ = 61, $n_g$ = 58, $n_h$ = 45 from at least three independent biological experiments). The Kolmogorov–Smirnov test was performed to test differences between conditions (ns $P$ > 0.05, **$P$ ≤ 0.01, ***$P$ ≤ 0.001).
Source data are available for this figure.

Krtcap2), and the mechanosensors Piezo1, Jup, and Dsp (Fig 5B and Table S1). Taken together, the proteomics data indicate that uniaxial cyclic stretching broadly alters the abundance of protein mechanosensors, including the autophagy-mediated depletion of several FA proteins. In addition, cyclic stretching leads to the accumulation of mitochondrial proteins in A7r5 ML cells.

**Uniaxial stretch promotes structural remodeling of focal adhesions**

To validate the proteomics data, we investigated by immunoblotting the protein levels of the structural FA proteins paxillin (Pxn) and vinculin (Vcl), as well as the chaperone Hsc70 (Hspa8) in

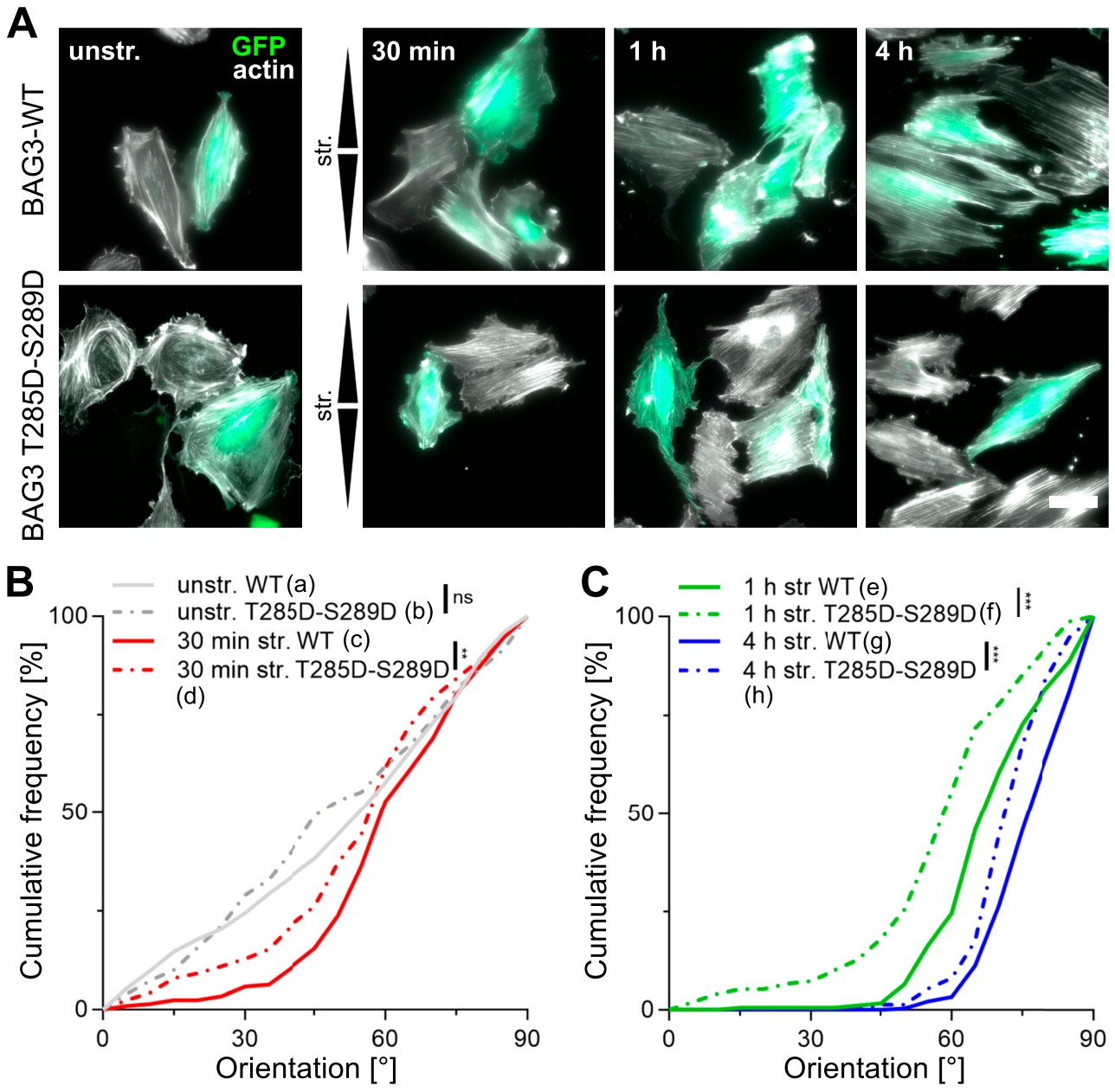

**Figure 4. Mechanoresponse of A7r5 SC cells is mediated by Bag3 (T285-S289) dephosphorylation.**
A7r5 subconfluent (SC) cells were cotransfected with GFP and human Bag3 (T285D-S289D) phosphosite mutant. The Bag3 mutant mimics the phosphorylation at T285 and S289 of Bag3, which prevents the execution of CASA. Results were compared with previously published data on A7r5 SC cells cotransfected with GFP and Bag3 WT plasmids (Lövenich et al, 2021). **(A)** Staining of actin fibers (white) in cotransfected A7r5 SC (green) after 30 min, and 1 and 4 h of uniaxial cyclic stretching (str.) and in unstretched control cells (unstr.) Arrowheads illustrate stretch direction. The scale bar is 50 $\mu m$. **(B, C)** Actin fiber orientation is represented as angular distribution from 0° to 90° in the stretch direction (0° means in stretch direction; 90° means perpendicular to stretch direction), plotted as cumulative frequencies observed in cotransfected A7r5 SC either in unstretched (unstr.) or after 30 min, and (C) 1 and 4 h of stretching (n represents the number of analyzed [transfected] cells; $n_a$ = 225, $n_b$ = 179, $n_c$ = 206, $n_d$ = 164, $n_e$ = 183, $n_f$ = 149, $n_g$ = 275, and $n_h$ = 75 from at least three independent biological experiments; data from Bag3 WT SC cells were taken from Lövenich et al [2021]). The Kolmogorov–Smirnov test was performed to test differences between conditions (ns $P > 0.05$, **$P \leq 0.01$, ***$P \leq 0.001$).
Source data are available for this figure.

A7r5 ML after 4 h of stretching (Fig S4). We observed consistent depletion of the FA protein paxillin (Fig S4A and C) and the chaperone Hsc70 (Hspa8) (Fig S4D and E) that is part of the CASA complex, but no depletion of vinculin (Fig S4A and B). To exclude transcriptional down-regulation, we quantified their mRNA levels by qRT–PCR (Fig S4F–H). We observed an increase in the mRNA levels of paxillin after 4 h of stretching, whereas those of vinculin and Hsc70 mRNAs remained unaltered compared with the unstretched controls. Next, we

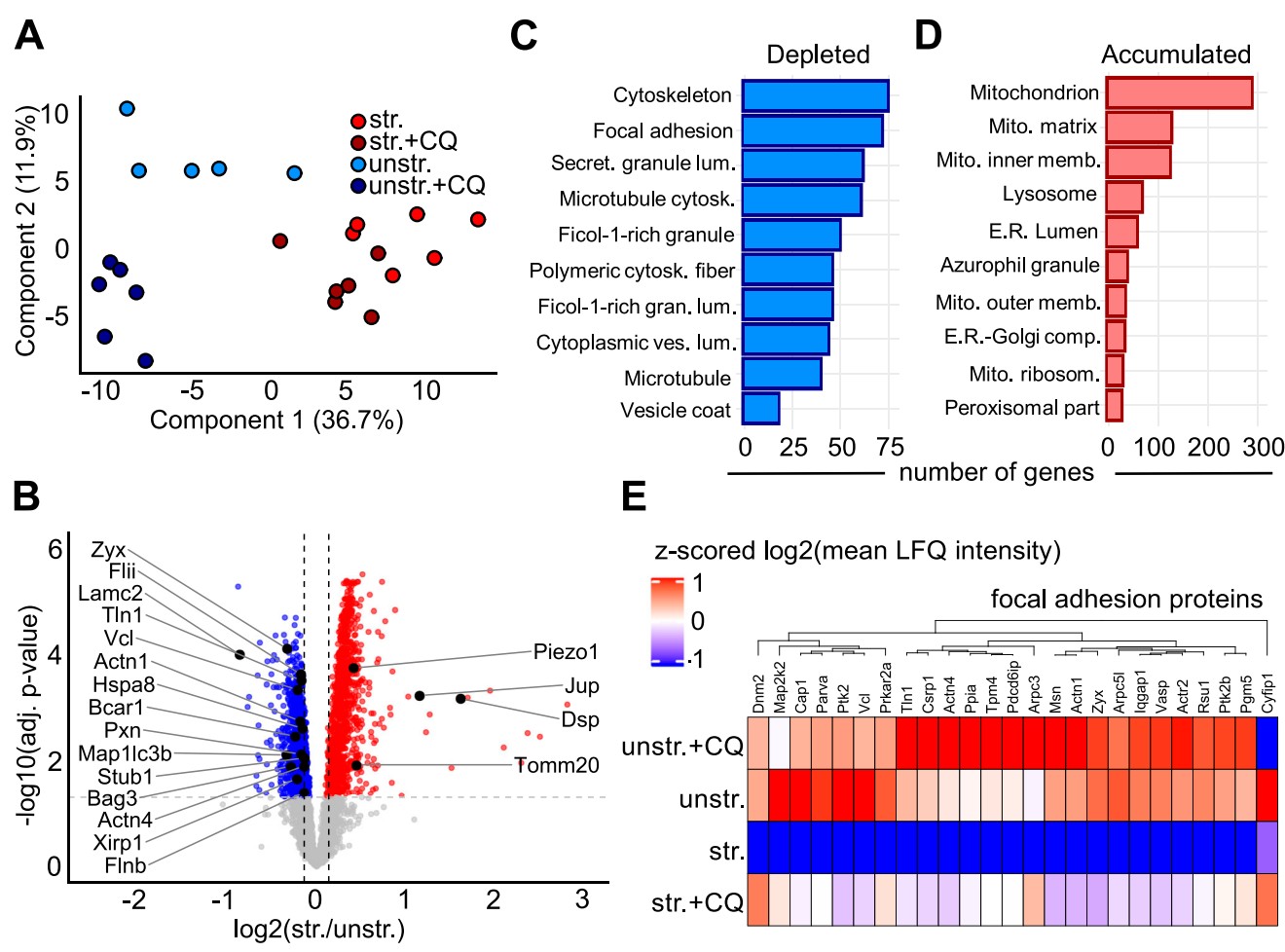

**Figure 5. Proteome adaptation in A7r5 monolayers after uniaxial cyclic stretching.**
Proteomic analysis of lysates from A7r5 monolayers (ML) after 4 h of uniaxial cyclic stretching (str.) and in unstretched control cells (unstr.), with chloroquine treatment (+CQ) or without. **(A)** Principal component analysis of 4,064 proteins. The first two principal components are displayed on the x- and y-axis. 4-h stretched cells (str.) replicate cluster away from unstretched control cells (unstr.) in the first component, which justifies 36.7% of the variance in the dataset. Chloroquine-treated cells (+CQ) are separated from untreated cells on the second component, which justifies 11.9% of the variance in the dataset. Chloroquine-treated cells (+CQ) are separated from untreated cells on the second component, which justifies 11.9% of the variance in the dataset. **(B)** Volcano plot comparing the log₂(str./unstr.) based on mean LFQ intensities of proteins in A7r5 ML after stretching (str.) and in unstretched control cells (unstr.) on the x-axis, and −log₁₀(adj. $P$-value) on the y-axis. Colors indicate accumulating (red) and depleted (blue) proteins after stretching (str. versus unstr., adj. $P$ <0.05). The LIMMA moderated $t$ test with the Benjamini–Hochberg false discovery rate control was used for statistical testing. **(C, D)** Functional enrichment analysis of significantly depleted and (D) accumulated proteins after stretching. Number of genes annotated in each category on the x-axis. All categories displayed were significantly enriched (FDR $P$ <0.05). **(E)** Heatmap displaying Z-score normalized mean log₂(LFQ intensity) of 25 focal adhesion proteins depleted in an autophagy-dependent manner (str. versus str.+CQ, $P$ <0.05) after stretching. Red indicates higher abundance, and blue indicates lower abundance. Experiments were repeated with n = 6 independent biological replicates.

investigated the effect of stretching on the structure of FAs, by co-immunostaining both paxillin and vinculin in A7r5 ML after 1 h or 4 h of uniaxial cyclic stretching (Fig 6A). Consistent with the proteomics (Fig 5B), the mean areas of the protein spots, for both proteins, were decreased after 1 and 4 h of stretching (Fig 6A, B, and D). Despite these changes, stretching induced no change in the fraction of FAs covered by each protein (Fig 6A, C, and E). Consequently, the mean areas of colocalized FAs decreased after both 1 and 4 h of stretching (Fig 6A and F), whereas their mean numbers were not affected (Fig 6A and G).

To determine whether the observed structural FA remodeling is involved in the strain-induced mechanoresponse, we further analyzed the orientation of FAs in A7r5 ML after 1 and 4 h of stretching (Fig 6H and I). Indeed, like actin fibers, FAs reoriented in a time-dependent manner

away from the strain direction. For visualization, upper and lower 95% confidence intervals (CI95) of FA reorientation for all stretching time points were plotted (Fig S5). In conclusion, the strain-induced mechanoresponse in A7r5 ML involves the structural remodeling and reorientation of FAs perpendicular to the strain direction, which includes a stretch-induced degradation of paxillin that is blocked by inhibiting autophagy.

## Mitochondrial activity supports the mechanoresponse to uniaxial stretch

We wondered whether the strain-induced increase in the abundance of mitochondrial proteins observed in the proteomics

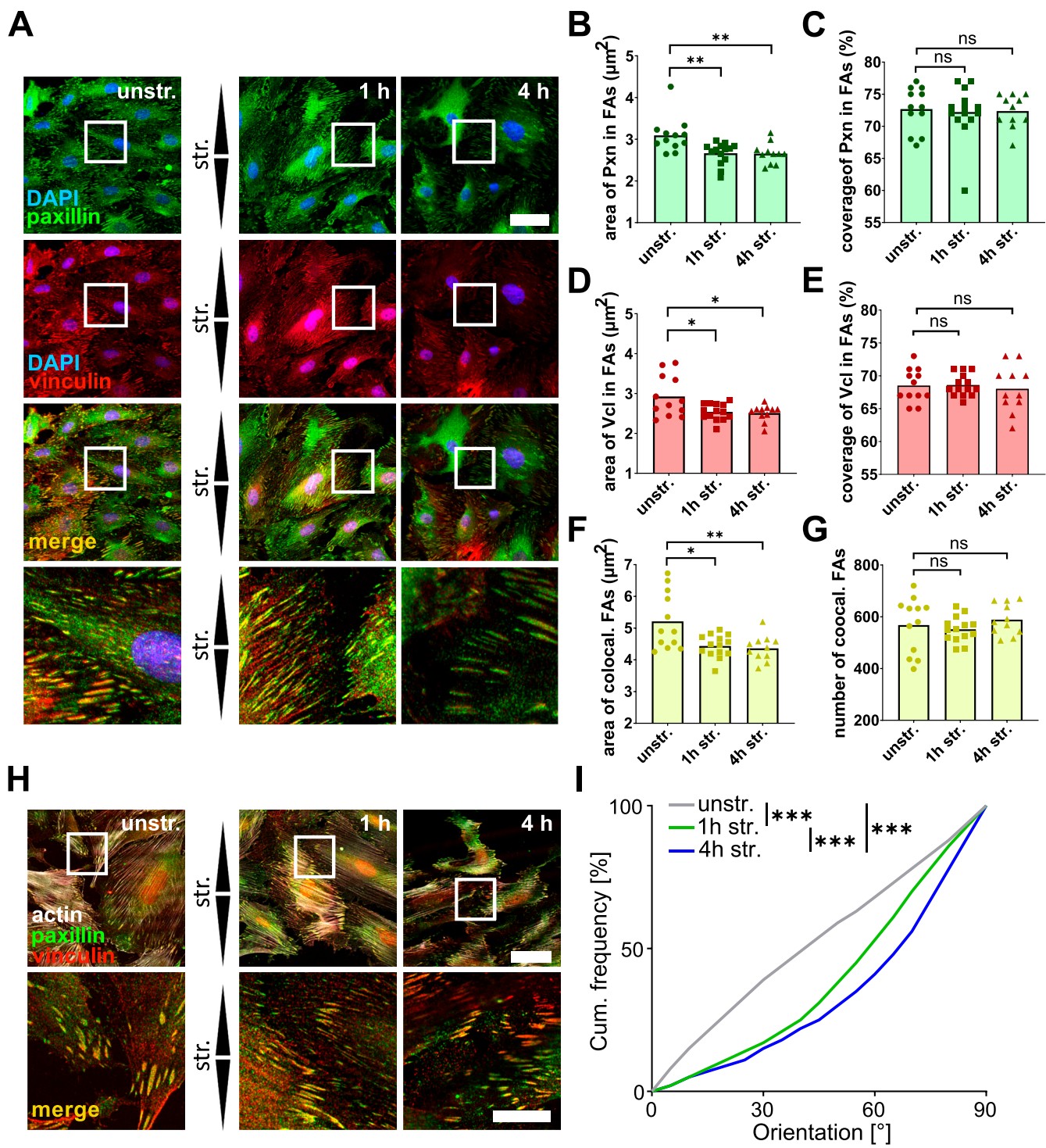

**Figure 6. Focal adhesion remodeling in A7r5 monolayer cells after uniaxial cyclic stretching.**
**(A)** Co-immunostaining of focal adhesion proteins paxillin (Pxn) (green) and vinculin (Vcl) (red) in A7r5 monolayer (ML) cells after 1 and 4 h of uniaxial cyclic stretching (str.) and in unstretched control cells (unstr.). Arrowheads illustrate stretch direction. The scale bar is 50 $\mu$m in overview and 20 $\mu$m in zoom images. **(B)** Mean area of Pxn spots ($\mu$m$^2$) in A7r5 ML after 1 and 4 h of stretching (str.) and in unstretched control cells (unstr.). **(C)** Fraction of FAs covered by Pxn in A7r5 ML after 1 and 4 h of stretching (str.) and in unstretched control cells (unstr.). **(D)** Mean area of Vcl spots ($\mu$m$^2$) in A7r5 ML after 1 and 4 h of stretching (str.) and unstretched control cells (unstr.). **(E)** Fraction of FAs covered by Vcl in A7r5 ML after 1 and 4 h of stretching (str.) and in unstretched control cells (unstr.). **(F)** Mean area of FAs ($\mu$m$^2$) in A7r5 ML after 1 and 4 h of stretching (str.) and in unstretched control cells (unstr.). **(G)** Mean number of FAs per cell in A7r5 ML after 1 and 4 h of stretching (str.) and in unstretched control cells (unstr.). Data are represented as mean and individual values of three individual biological experiments (n represents the number of analyzed images, with typically >10 cells each $n_{unstr.}$ = 13, $n_{1\ h\ str.}$ = 18, $n_{4\ h\ str.}$ = 12 resulting in at least 120 analyzed cells per group). Results marked based on significance were analyzed by a $t$ test (ns $P$ > 0.05, *$P$ ≤ 0.05, **$P$ ≤ 0.01). **(H)** Co-immunostaining of focal adhesion proteins paxillin (Pxn) (green) and vinculin (Vcl) (red) in A7r5 monolayers (ML) after 1 and 4 h of uniaxial cyclic stretching (str.) and in unstretched control cells (unstr.). Arrowheads illustrate stretch direction. The scale bar is 50 $\mu$m in overview and 20 $\mu$m in zoom

experiment (Figs 5D and S6A) was associated with changes in mitochondrial abundance and/or mitochondrial activity. To investigate this, the mitochondrial outer membrane protein Tomm20 (translocase of outer mitochondrial membrane 20), which accumulated in A7r5 ML after 4 h of stretching (Figs 5B and S6B and C), was selected as a proxy for mitochondrial abundance and imaged after stretching. There was no change in the mean area or number of mitochondria per cell after stretching (Fig S6D, E, and F). We then inspected mitochondrial mass using the MitoTracker Green FM staining and observed an increase in staining intensity after 4 h of stretching, but not after 1 h (Fig S6G and H). Because we did not observe differences in the abundance of mitochondria, but an increase in mitochondrial mass after 4 h of stretching, this suggested that the mitochondrial activity may have increased after prolonged stretching. To test this, we compared the mitochondrial membrane potential of unstretched cells and after 4 h of stretching using the JC-1 assay (Fig 7A–C). The JC-1 dye accumulates in mitochondria in a membrane potential ($Δψm$)–dependent manner. In mitochondria with low $Δψm$, it remains in its monomeric form and emits green fluorescence, whereas in mitochondria with high $Δψm$, it aggregates and emits red fluorescence (Fig 7A). Carbonyl cyanide m-chlorophenylhydrazone (cccp) was used here as a control for mitochondrial depolarization. We observed that the mitochondrial membrane potential was increased after 4 h of stretching (Fig 7B and C). To explore this further, we assessed the cellular levels of ATP in A7r5 ML after 1 and 4 h of stretching and found no changes in the ATP levels (Fig S7A). To understand whether stretching induces transcription of genes related to the mitochondrial proteome, we measured the transcript levels of 84 proteins associated with mitochondria in A7r5 ML after 4 h of stretching (Fig S7B). We observed increased expression ($\log_2$(str./unstr.) > 0.2) of 36 mitochondrial transcripts, whereas only seven transcripts showed decreased expression ($\log_2$(str./unstr.) < −0.2), and 41 transcripts did not change after stretching. Taken together, these results suggest that mitochondria were activated in response to uniaxial cyclic stretching.

Finally, we investigated the connection between strain-induced mitochondrial activation and adaptive reorientation. To do so, we examined the actin fiber orientation in A7r5 ML after 1 and 4 h of uniaxial cyclic stretching, with or without treatment (untreated) with the mitochondrial ATP synthase inhibitor oligomycin (+oligomycin) (Fig 7D and E). In unstretched cells, the oligomycin treatment had no effect on the random actin fiber orientation. Upon stretching, treatment with oligomycin slowed the actin fiber reorientation after both 1 and 4 h of stretching. For visualization, upper and lower 95% confidence intervals (CI95) of untreated and oligomycin-treated cells for all stretching time points were plotted (Fig S7C). In conclusion, these results suggest that uniaxial cyclic stretching activates mitochondria in A7r5 ML cells to maintain the cellular levels of ATP and support the strain-induced

mechanoresponse and facilitate actin fiber reorientation perpendicular to the stretch direction.

## Discussion

Uniaxial cyclic stretching promotes actin fiber reorientation in smooth muscle cells in vitro, which helps alleviate cellular stress by minimizing cell elongation in the strain direction (Livne et al, 2014). In SC cells, this is accompanied by a transient increase in APs and involves the disposal of damaged mechanosensors via CASA (Ulbricht et al, 2013; Lövenich et al, 2021).

Here, we investigated the strain-induced mechanoresponse in A7r5 rat smooth muscle monolayer (ML) cells, which form additional mechanosensory structures, such as tight junctions (Citi, 2019) and AJs (Noethel et al, 2018). Despite the presence of cell–cell contacts, reorientation of the actin fibers perpendicular to the strain direction upon stretching occurred on the same timescale as in subconfluent (SC) cells. Moreover, we demonstrate here that monolayers respond in a similar way as subconfluent cells to cyclic stretching (Lövenich et al, 2021). The ML mechanoresponse is characterized by a transient accumulation of APs (Fig 2), and inhibiting autophagy, through either chemical inhibition (Fig 3) or genetic manipulation of the CASA regulator Bag3 (Fig 4), slowed the adaptive reorientation of the cytoskeleton. These findings reinforce the notion that autophagy, especially Bag3-mediated autophagy, acts as a potential mechanosensitive mechanism for efficient cellular adaptation to mechanical stress.

The prominent role of autophagy in the phenotypic cellular mechanoresponse prompted us to investigate the proteome of ML cells in response to stretching and to assess whether inhibiting autophagy would prevent the degradation of mechanosensors. For this, we selected the 4-h time point, when cells had fully adapted, and the number of APs had returned to basal levels. Stretching had a more prominent effect on the proteome than inhibiting autophagy. This was consistent with the observation that the adaptive reorientation was slowed, rather than completely impaired, with CQ treatment (Fig 3). Stretching also induced the depletion of numerous cytoskeletal and cell adhesion proteins including focal adhesions (Fig 5) (Table S1). The depletion of the FA protein Pxn after stretching was verified by imaging and could be extended by Vcl, revealing a reduction in the size of FAs, but not in their number. Interestingly, the remodeled FAs reoriented perpendicular to the strain direction (Fig 6H and I), confirming observations in other cell types and supporting a model (De, 2018) in which FAs reorient away from the strain direction as part of the mechanoresponse to cyclic stretching and along the actin reorientation. Consistent with the essential role of CASA during mechanical stress, whereby certain regulators turn over alongside their targets (Ulbricht et al, 2013;

---

images. **(I)** FA reorientation is represented as an angular distribution from 0° to 90° in the stretch direction (0° means in stretch direction; 90° means perpendicular to stretch direction), plotted as cumulative frequencies observed in A7r5 ML after 1 and 4 h of stretching (n represents the number of analyzed FA spots, $n_{unstr.}$ = 5,140, $n_{1\ h\ str.}$ = 4,012, $n_{4\ h\ str.}$ = 5,620; for each group, a minimum of 100 cells have been analyzed). The Kolmogorov–Smirnov test was performed to test differences between conditions (***$P$ ≤ 0.001).

Source data are available for this figure.

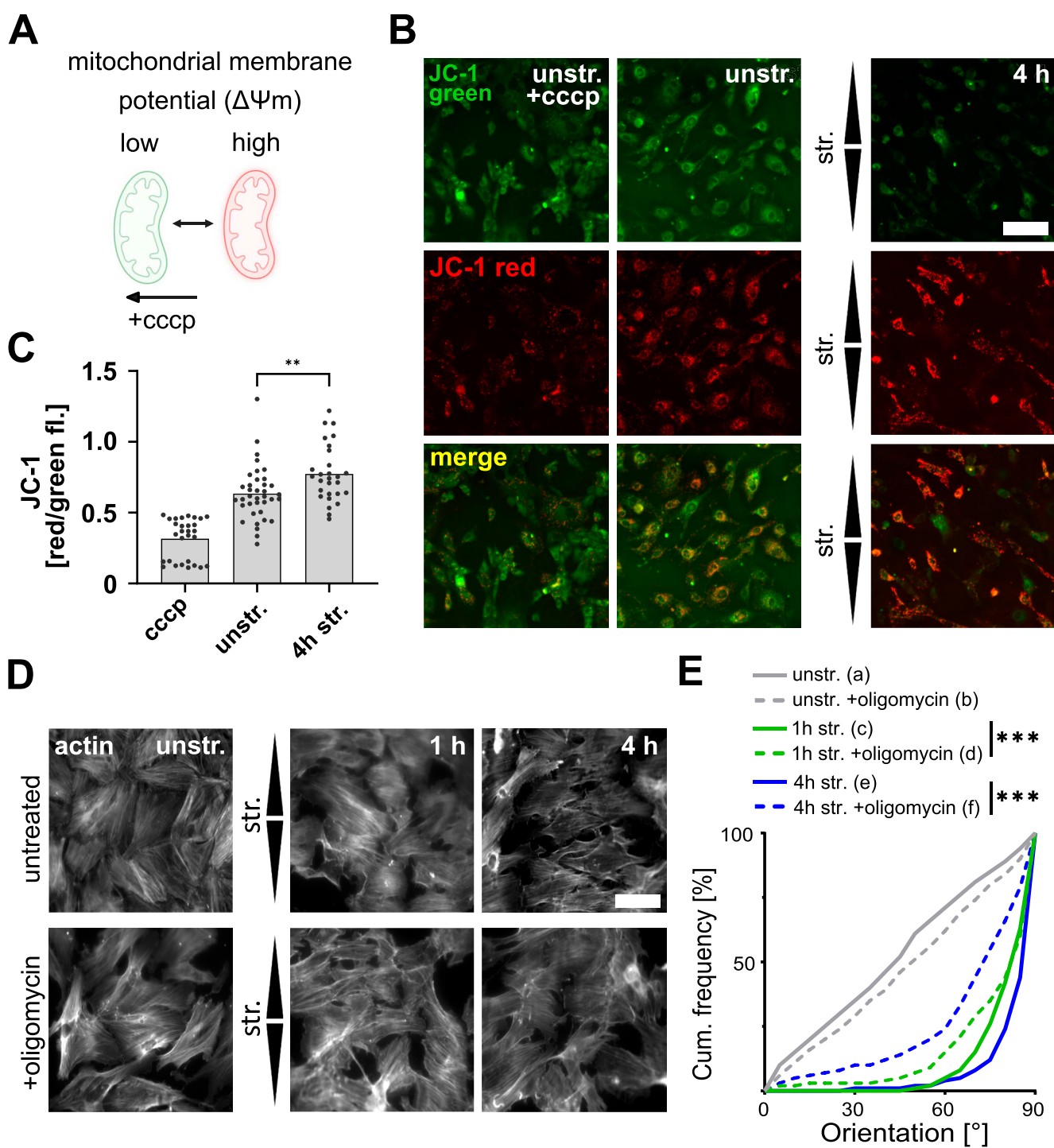

**Figure 7. Mechanoresponse of A7r5 monolayer cells is mediated by increased mitochondrial membrane potential.**
**(A)** Scheme of the JC-1 assay to analyze mitochondrial membrane potential (Δψm). **(B)** JC-1 staining of A7r5 monolayer cells after 4 h of uniaxial cyclic stretching (str.), in unstretched (unstr.) control cells, and in unstretched control cells treated for 5 min with cccp (unstr. +cccp). Green fluorescent signals represent apoptotic and dead cells, whereas intact mitochondria in viable cells show red fluorescence signal. The scale bar is 50 $\mu$m. Stretch direction is indicated by arrowheads. **(C)** Analysis of JC-1 by illustrating the ratio of red to green fluorescent signal. Data are represented as mean and individual values (n represents the number of analyzed images from three biological individual experiments; $n_{cccp}$ = 29, $n_{unstr.}$ = 37, and $n_{4\ h\ str.}$ = 28). Test results marked based on significance were analyzed by a t test (**$P \leq 0.01$). **(D)** Staining of actin fibers (white) in A7r5 monolayers (ML) after 30 min, and 1 and 4 h of uniaxial cyclic stretching (str.) and in unstretched control cells (unstr.), with oligomycin treatment (+oligomycin) or without. Arrowheads illustrate stretch direction. The scale bar is 50 $\mu$m. **(E)** Actin fiber orientation represented as angular distribution from 0° to 90° in the stretch direction (0° means in stretch direction; 90° means perpendicular to stretch direction), plotted as cumulative frequencies observed in A7r5 ML after 30 min, and 1 and 4 h of stretching, and in unstretched control cells (unstr.), with oligomycin treatment or without (n represents the number of cells; $n_a$ = 726, $n_b$ = 789,

Ottensmeyer et al, 2024), Bag3, Hspa8, Map1lc3b, and others were also found depleted after stretching in this study using mass spectrometry. In addition, impairing CASA by overexpressing the dominant negative mutant (Ottensmeyer et al, 2024) Bag3 (T285D-S289D) slowed the adaptive reorientation upon stretching, uncovering a new functional cellular mechanism that is mediated by Bag3 dephosphorylation. The observed stretch-induced autophagic degradation of the FA protein Pxn might potentially involve Bag3 and other CASA components. However, this hypothesis still requires further validation. In any case, the observed increase of paxillin mRNA levels during stretching points to a potential mechanosensitive mechanism for cellular adaptation to mechanical stress through dynamic FA degradation and reassembly.

Strikingly, mitochondrial proteins were the largest group among those accumulated after stretching. Imaging of mitochondria revealed that although their abundance measured by area and number was not altered, their mass and the membrane potential were increased. The elevated mitochondrial protein and transcript levels, along with an increased mitochondrial membrane potential, suggest that mitochondria were activated in response to stretching. However, no changes in cellular ATP were observed. This may be explained by a balanced production and consumption of ATP, where increased ATP synthesis would support actin dynamics such as polymerization and reorientation (Kanematsu et al, 2022), estimated to consume up to 50% of cellular ATP (DeWane et al, 2021). The disassembly of stress fibers occurring through reduced substrate stiffness down-regulates metabolic enzymes for glycolysis stating a direct mechanism of energy consumption and the actomyosin cytoskeleton (Park et al, 2020), which must be positively regulated for even stronger mechanical stimuli during stretching. Mitochondrial activation likely helps cells to meet the increased energetic demands of reorientation and maintain homeostasis under strain. In line with this, previous studies suggested that mechanical stimuli can modulate mitochondrial function (Picard et al, 2013; Wolff et al, 2013; MacVicar & Langer, 2021). This includes the regulation of mitochondrial activity by the mechanosensor Piezo1 (Li et al, 2022), which accumulated after stretching in our study investigated by mass spectrometry. Consistent with this, stretch-induced calcium influx into trabecular meshwork cells was shown to be regulated by Piezo (Uchida et al, 2021). Stretch-induced calcium influx is taken up via the voltage-dependent anion channel isoform 1 (VDAC1) and the mitochondrial calcium uniporter (MCU) at the inner mitochondrial membrane and further acts as a secondary messenger to activate TCA cycle enzymes and the ETC complexes I and II (Chaanine, 2021). This may also contribute to the mitochondrial activation we observed here. Stretching of smooth muscle cells has also been reported to induce transcription of endoplasmic reticulum (ER) stress–related genes. ER stress is associated with increased mitochondrial metabolism depending on organelle coupling and calcium transfer (Bravo et al, 2012). An alternative explanation for the increased abundance of mitochondrial proteins and biomass could also be the impairment of mitophagy after stretching. Previously, mitochondrial protein levels were found altered in the hearts of mice expressing the Bag3 (P209L) loss-of-function mutant, unable to execute CASA (Kimura et al, 2021). Interestingly, Bag3 dephosphorylation at residues 285/289 promotes its interaction with multiple mitochondrial proteins (Ottensmeyer et al, 2024). In addition, autophagosome formation during CASA depends on the dephosphorylation-dependent interaction of Bag3 with Rab7a, a Rab GTPase involved in the early steps of mitophagy (Heo et al, 2018; Yan et al, 2022). Bag3 dephosphorylation itself is triggered by chemical induction of mitophagy (Ottensmeyer et al, 2024), and potentially underlines its recruitment to depolarized mitochondria (Tahrir et al, 2017). Therefore, it appears plausible that the recruitment of Bag3 to damaged cellular structures, including focal adhesions, reduces its availability to participate in mitophagy, resulting in an accumulation of mitochondrial proteins and ultimately contributing to mitochondrial activation.

In conclusion, this study identifies dynamic FA degradation and reassembly as an important part of the cellular response to mechanical strain in cell monolayers, apparently driven by transient autophagosome formation and mitochondrial activation, and suggests the Bag3-mediated proteostasis system as a critical mediator of the cellular responses to mechanical stress, being able to adjust autophagy and mitochondrial function.

# Materials and Methods

### Cell culture

Rat smooth muscle cells (A7r5) from the American Type Culture Collection (ATCC) were grown at 37°C and 5% (vol/vol) $CO_2$ in a humidified atmosphere. Cells were grown in DMEM/low-glucose/pyruvate/no glutamine, no phenol red, medium (Life Technologies) supplemented with 10% (vol/vol) FBS Premium (P30-3302; Pan Biotech), 1% (vol/vol) penicillin/streptomycin (Sigma-Aldrich), and 2 mM GlutaMAX (Thermo Fisher Scientific).

### Elastomeric substrate preparation

Elastic polydimethylsiloxane (PDMS) chambers were prepared as before (Faust et al, 2011). To mimic natural elastic conditions, experiments were executed on 50 kPa elastomeric substrates, coated with human fibronectin (FN). The coating was performed with 20 $\mu$g/ml FN (Corning) in PBS for 3 h at 37°C before seeding the cells. A7r5 cells were seeded 24 h before stretching at a density of 12,500 cells/cm$^2$ to generate monolayers.

$n_c$ = 638, $n_d$ = 466, $n_e$ = 690, and $n_f$ = 616, measured from three independent biological experiments). The Kolmogorov–Smirnov test was performed to test differences between conditions (ns $P > 0.05$, *$P \leq 0.05$, **$P \leq 0.01$, ***$P \leq 0.001$).
Source data are available for this figure.

## Cell stretching

Before stretching, cells were either treated for 1 h with 100 $\mu$M chloroquine diphosphate salt (CQ) (Sigma-Aldrich) or incubated in growth media. Uniaxial cyclic stretching was performed in six elastomeric chambers in parallel (Noethel et al, 2018), using physiology mimicking parameters (20% amplitude, 300 mHz frequency), as before (Lövenich et al, 2021). For experiments with human Bag3, cells were cotransfected with human Bag3-WT or Bag3-T285D/S289D and GFP plasmid DNA using Lipofectamine 3000 (Thermo Fisher Scientific), according to the manufacturer's protocol, 24 h after seeding on the chambers with 500 ng DNA and stretched after 24 h. All plasmids used were published before (Kathage et al, 2017; Ottensmeyer et al, 2024).

## Immunocytochemistry

Immunocytochemistry was performed as before (Lövenich et al, 2021). In short, cells were fixed either in 100% ice-cold methanol (10 min at −20°C) for LC3B (Map1lc3b) immunostainings or in 3.7% (vol/vol) paraformaldehyde (PFA) (Sigma-Aldrich) in cytoskeleton buffer (CB) (150 mM NaCl, 5 mM MgCl$_2$, 5 mM EGTA, 5 mM glucose, 10 mM 2-(N-morpholino)ethanesulfonic acid, pH 6.1) for all other immunostainings (15 min, 37°C). Fixed cells were washed twice at RT with CB, and PFA fixed cells in 100 mM glycine in CB after all incubations, unless indicated otherwise. Next, membranes were permeabilized (10 min, RT) with 0.05% (vol/vol) Triton X-100 (Sigma-Aldrich) in CB, washed three times, and blocked (1 h, RT) in 5% milk powder (Roth) in CB. After blocking, cells were incubated (16 h, 4°C) with primary antibodies, all at 0.2% (vol/vol) in 1% milk powder in CB (rabbit anti-LC3B [Map1lc3b] [2775; Cell Signaling], mouse anti-paxillin [Pxn] [AHO0492; Invitrogen], rabbit anti-$\alpha$-catenin [Ctnna1] [C2081; Sigma-Aldrich], and anti-Tomm20 [sc-17764; Santa Cruz]). After washing three times, cells were incubated (1 h, RT) with 0.2% (vol/vol) goat anti-rabbit (A11034; Life Technologies) or 0.2% (vol/vol) donkey anti-mouse (A21202; Invitrogen) secondary antibodies in 0.1% 4',6-diamidino-2-phenylindole (DAPI) (Invitrogen) and 1% milk powder in CB. If indicated, actin fibers were simultaneously stained with 0.2% (vol/vol) label of Alexa Fluor 546 phalloidin (A22283; Invitrogen) during secondary antibody incubation. For co-immunostaining of paxillin (Pxn) and vinculin (Vcl), after fixation and blocking, cells were incubated overnight with both mouse anti-paxillin (Pxn) (AHO0492; Invitrogen) and rabbit anti-vinculin (Vcl) (700062; Invitrogen) antibodies. After washing, cells were incubated with donkey anti-mouse (A21202; Invitrogen) and goat anti-rabbit (A21071; Life Technologies) secondary antibodies in 1% milk powder. After washing three times, and once with deionized H$_2$O, Fluoromount Aqueous Mounting Medium (Sigma-Aldrich) and a glass coverslip were added on top. After this, the chambers were stuck on a micro slide (VWR) and dried overnight before microscopy.

## Microscopy

Paxillin (Pxn), vinculin (Vcl), $\alpha$-catenin (Ctnna1), LC3B (Map1lc3b), actin, and Tomm20 immunostainings were analyzed on a confocal laser scanning microscopy (LSM 880; Carl Zeiss) using Airyscan mode with a 40x EC-Plan-Neofluar/ph3/1.3 NA oil objective (Carl Zeiss) or a 63x Plan Apochromat/1.4 NA oil objective (Carl Zeiss). Images were processed with Zen Black software (Carl Zeiss). Z-stacks of cells were performed using appropriate settings for excitation and emission depending on the secondary antibodies used. For Pxn, Ctnna1, and Tomm20 immunostainings, Alexa Fluor 488 signals were detected using a BP 495- to 550-nm filter, and for Vcl immunostainings, Alexa Fluor 647 signals were detected using a 633-nm laser, and a BP 420- to 480-nm filter for DAPI and a BP 570- to 620-nm filter for Alexa Fluor 546 were used to detect the actin signal. For LC3B immunostainings, Alexa Fluor 488 was detected using a BP 495- to 620-nm filter supported by a BP 420- to 480-nm filter for DAPI. For imaging actin fibers, cells were imaged with a widefield microscope (AxioObserver; Carl Zeiss) equipped with a Colibri 7 (Carl Zeiss) light source, an Axiocam 712 mono camera (Carl Zeiss), and a 40x EC-Plan-Neofluar/ph3/1.3 NA oil objective (Carl Zeiss), using a BP 570- to 620-nm filter for detecting Alexa Fluor 546 and a BP 495- to 550-nm filter for BagAG3-GFP cotransfection experiments.

## LC3B-spot analysis

Cells with autophagy inhibition have been pretreated for 1 h with 100 $\mu$m chloroquine diphosphate salt (CQ) (C6628; Sigma-Aldrich) before starting the experiment, and it was left on the cells during the experiment. LC3B (Map1lc3b)-spot quantification in A7r5 cells was performed as before (Lövenich et al, 2021) but without setting a cell mask aiming to analyze LC3B spots in the whole monolayer in each image. In a first step, an intensity threshold was selected to separate LC3B spots from the background signal. All spots with an area <70 pixels (pixel size 0.071 $\mu$m) were rejected. To separate big clusters of spots, the watershed algorithm was used. Therefore, the distance transformation of each spot was applied to the segmented image and local maxima are used as starting points for watershedding. As the cell monolayers overlay the whole image, cell areas could not be detected for normalization through the fluorescent signals. Therefore, the number of remaining spots was divided by the number of manually counted nuclei per image (DAPI) (typically 3–8 cells per image) to normalize the number of LC3B spots to the cell numbers, and also to avoid the misinterpretation of potentially decreased cell areas because of CQ treatment. The Mann–Whitney test was performed to test differences between conditions (ns $P > 0.05$, *$P \leq 0.05$, **$P \leq 0.01$, ***$P \leq 0.001$).

## Analysis of actin fiber orientation

The orientation of actin fibers was analyzed as described before (Faust et al, 2011). As single cells within the monolayer could not be marked as before this, the number of cells per image was calculated by a DAPI staining and counting the nuclei per image (typically >15 nuclei per image). For the actin fiber orientation, a grid with the mean number of cells per image (5 × 3 cells) was laid on the image (353 × 259 $\mu$m), and each square was processed as a cell equivalent for the reorientation analysis. Actin fiber orientations were determined from gray value gradients. The most frequent orientation (i.e., the maximum of the distribution) was taken

as the main actin fiber orientation of a given square. Actin fiber orientations were plotted as cumulative distributions from parallel (0°) to perpendicular (90°) to the strain direction for illustration. For the statistical analysis of actin fiber reorientation, mean actin orientations of each image were calculated. Because of communication of cell units within one image, only these averaged values were used as independent measurements (n). For Bag3 experiments, only cells cotransfected with GFP, and the respective Bag3 versions were analyzed for actin fiber orientation, by defining them manually based on green fluorescence. The Kolmogorov–Smirnov test was performed to test differences between conditions (ns $P > 0.05$, *$P \leq 0.05$, **$P \leq 0.01$, ***$P \leq 0.001$). Experiments were performed at least three times independently, otherwise it is indicated.

### Focal adhesion colocalization and reorientation analysis

The software to detect focal adhesions and determine the colocalization between focal adhesion proteins in different image channels was written in Python 3.11.7. In short, the sequence of logical steps is as follows: first, the cell-covered area is identified; second, within this area, spots of high intensity were localized in both channels (vinculin and paxillin); third, only spots with a minimum overlap with spots in the other channel were retained; and finally, the total area of both overlapping spots in the two channels was defined as "focal adhesion" (cf. also Fig S1G). The cell-covered area was determined from the paxillin channel. First, the local contrast was enhanced by means of CLAHE with a clip limit of 2%. In the next step, a median filter with a disk-shaped structure element with a radius of five pixels (0.183 $\mu$m/pixel) removed salt-and-pepper noise, and then, a Gaussian filter (sigma five pixels) smoothed the image. From this image, a binary mask was generated. The necessary intensity threshold was determined from the smoothed (Savitzky–Golay filter, 21 points cubic polynomial) histogram of gray values in which the first local maximum was searched. The intensity value at 0.2 full widths at half maximum to the right of this peak was used as a threshold. Only pixels within this mask were considered in the following. To identify spots in a given channel, the images were first smoothed by a Gaussian filter with a sigma of three pixels and the contrast was enhanced by CLAHE with a clip level of 2%. For each pixel, the local z-score was calculated in a square environment of 35 pixels. A z-score of 1 was used as a threshold for creating binary masks. Only connected spots whose area exceeded 100 pixels were retained. In the next step, spots in both channels were compared. Only spots that had an overlap with a spot in the other channel of at least five pixels were retained. In the last step, the union of both connected spots was defined as focal adhesion. Orientation of FAs, and areas of spots and FAs were calculated using the skimage function regionprops (https://scikit-image.org/docs/stable/api/skimage.measure.html#skimage.measure.regionprops). FA orientations were plotted as cumulative distributions from parallel (0°) to perpendicular (90°) to the strain direction. The Kolmogorov–Smirnov test was performed to test differences between conditions (ns $P > 0.05$, *$P \leq 0.05$, **$P \leq 0.01$, ***$P \leq 0.001$). Experiments were performed at least three times biologically independently.

### Analysis of Tomm20 immunostaining

The program to analyze mitochondria in each single plane (z-direction) of the image was written in Python version 3.11.7. First, a local median filter with a disk shape of two pixels (0.183 $\mu$m/pixel) was performed to smooth the image. Then, a top-hat transformation was used with a disk shape of five pixels for the opening operation. Next, contrast limited adaptive histogram equalization (CLAHE) (Zuiderveld, 1994) with a clip limit of 2% was used to enhance the local contrast. By default, kernel size is 1/8 of the image height by 1/8 of its width. The local image thresholding method (radius 15 pixels) as developed by Phansalkar et al (2011) was performed to finally binarize mitochondria. Overlapping objects were separated by a binary opening operation with a disk shape of two pixels. Then, the binary objects were skeletonized. Mitochondrial average area and number were calculated based on this skeleton. This procedure was performed for each plane in the image stack. Mitochondrial numbers per cell were extracted and averaged for each image (analyzed as z-stack of manually marked cell masks by the actin signal). The number of mitochondria was extracted and divided by the number of cells in each image. Mitochondrial area was extracted by averaging all mitochondrial binary objects and normalized to the total cell area in each image, based on the actin staining.

### MitoTracker staining, imaging, and analysis

After stretching, and respective unstretched control cells, culture media were replaced with the MitoTracker Green FM (M7514; Invitrogen) staining solution (100 nM dye in culture media), and chambers were incubated for 30 min at 37°C. Cells were washed three times with PBS, and normal culture medium was added for imaging. As no differences in mitochondrial size and number were detected, the MitoTracker was used as an endpoint assay to detect mitochondrial membrane potential, which accumulates in the mitochondrial intermembrane space in a pH-dependent manner. Mitochondrial staining (MitoTracker) was analyzed using an upright microscope (Axio Imager.M2m by Zeiss). To detect the fluorescence intensity, a wavelength of 493 nm was used to excite the dye. The 38 HE Green fluorescent protein filter (Zeiss) was used to separate fluorescence and excitation. Ten images were taken per chamber, as stacks along the z-axis, with an LD Plan-Neofluar 40x/0.6 Korr Ph2 M27.

Analysis of mitochondria was conducted using a Python-based program. To analyze the average gray value of mitochondria, a maximum-intensity projection (MIP) of the green fluorescence signal was performed and a cell mask was calculated. The background of the image was corrected by subtracting the morphologically opened image (using a squared structured element with a size of 251 × 251 pixels, pixel size 0.18 $\mu$m), from the MIP image. From this background-corrected image, the minimum gray value was subtracted and a threshold for segmentation was calculated by Otsu's method (Liu & Yu, 2009). To also include darker pixels in the cell mask, this threshold was divided by three. After segmentation, the cell mask was post-processed using morphological operations

(opening, closing) and by removing objects with areas below 1,000 pixels. All gray values within this cell mask were averaged as the mean cell intensity. All gray values outside of the cell mask were averaged as the mean background intensity. Mitochondrial intensity was then defined as mean cell intensity subtracted by the background intensity. Mitochondrial intensities were normalized by scaling the average value of the unstretched cells to 1, and all other values are related to that.

### JC-1 staining, imaging, and analysis

After stretching, and respective unstretched control cells, medium exchange and one time washing cells with PBS, samples were incubated with JC-1 dye (2 $\mu$M, 15–30 min, 37°C, 5% $CO_2$). Positive controls received carbonyl cyanide 3-chlorophenylhydrazone (cccp) (50 $\mu$M, 5 min). After washing, samples were maintained in PBS, protected from light, and immediately assessed for fluorescence. Fluorescence microscopy was performed using a dual-band-pass filter. JC-1 emitted green fluorescence at ~530 nm in apoptotic/dead cells (monomeric form), whereas intact mitochondria in viable cells showed red signals at ~590 nm (dye aggregates). Image evaluation focused on the red/green ratio. Microscopy was applied, with 10 representative images typically acquired per sample. For image analysis, red and green fluorescence signals were analyzed by identifying bright regions within the images. Maximum-intensity projections of both channels were generated, and an additional composite image was obtained by summing the two MIPs. The summed image was smoothed using a Gaussian filter ($\sigma$ = 3, pixel size = 0.721 $\mu$m) and subsequently z-score–normalized (Hersch et al, 2013). A morphological opening was computed from the normalized image and subtracted to enhance local features. The resulting image was segmented using a fixed threshold of 0.6.

### Cell lysis and preparation for mass spectrometry

Cells were washed twice with warm PBS and lysed by adding 100 mM 4-(2-hydroxyethyl)-1-piperazineethanesulfonic acid (Hepes), 1% SDS buffer supplemented with protease inhibitor cocktail (Sigma-Aldrich), and scratching cells with a scraper. Complete cell lysis and protein denaturation were performed by three rounds of heating (5 min, 90°C, shaking) and cooling (5 min, ice). Protein concentration was estimated using Pierce BCA (bicinchoninic acid assay) Protein Assay Kit (Thermo Fisher Scientific), and 20 $\mu$g of protein per sample was reduced with 10 mM dithiothreitol (DTT) (30 min, 37°C, shaking) and alkylated with 50 mM chloroacetamide (CAA) (30 min, RT) in the dark. Alkylation was quenched with 50 mM DTT (20 min, shaking) before protein purification using paramagnetic SP3 beads (Cytiva Europe) in 80% ethanol (20 min, shaking). The beads were washed twice with 80% acetonitrile, resuspended in 50 mM Hepes, pH 7.4, 5 mM $CaCl_2$ buffer added with trypsin (Serva) in a 1:100 (wt/wt) protease-to-proteome ratio, and digested overnight (18 h, 37°C, shaking). Peptides were desalted with self-packed C18 STAGE tips and resuspended in 0.1% formic acid.

### Mass spectrometry data acquisition

An estimated 1 $\mu$g desalted peptides per sample were loaded on an Ultimate 3000 RSLC nano chromatography system (Thermo Fisher Scientific) with a $\mu$PAC reverse-phase trap column (Pharma-Fluidics) and a 50-cm $\mu$PAC reverse-phase analytical column (PharmaFluidics) operated at 40°C. Peptides were eluted using a 600 nl/min flow rate and a binary gradient from 1% to 30% eluent B (A: 0.1% formic acid in water, B: 0.1% formic acid in acetonitrile) with a 90-min separation time, in a 2-h run per sample. Separated peptides were ionized using a Captive Spray ion source (Bruker) and introduced into an Impact II Q-TOF mass spectrometer (Bruker), as previously described (Mooney et al, 2024). Briefly, data were acquired with HyStar software (Bruker) using a data-independent acquisition (DIA) method with 400–1,200 precursor mass range for MS1, with 32 windows of 25 m/z size with 0.5 m/z overlap, and 200–1,750 for MS2 fragment spectrum acquisition at 15 Hz.

### Proteome data analysis

DIA raw files were searched with FragPipe (Yu et al, 2023) (version 21) using the default workflow (DIA_Speclib_Quant), with the following settings: spectral library generation; cysteine (C) carbamidomethylation set as fixed modification; methionine (M) excision (at protein N-terminal) and oxidation set as variable modifications; Max. number of variable modifications per peptide set to 3; trypsin set as digestion enzyme and max. number of missed cleavages 2; a FASTA file containing UniProt rat canonical protein sequences and isoforms (UP000002494_10116, downloaded on May 2023) added with common contaminants and reversed decoys was used for the search. Control unstretched, untreated (unstr. 2), was not included in the analysis because of significantly lower identifications compared with other samples. Quantitative data were preprocessed by excluding contaminants (non-rat proteins). The LFQ intensities of protein groups containing 100% valid values in all samples (4,064 proteins) were $log_2$-transformed, and differential expression analysis was performed in R (V.4.02) with the limma package (Ritchie et al, 2015). The Benjamini–Hochberg method was used for multiple hypothesis testing. Functional analysis was performed using the EnrichR package (Xie et al, 2021). The volcano plot and functional analysis bar plots were drawn using the Ggplot package. The heatmap was plotted using the ComplexHeatmaps package (Gu et al, 2016). Visualization of pathway enrichment was performed by plotting $log_2$ fold change ratios in the respective KEGG pathways using Pathview (Luo & Brouwer, 2013).

### Crude protein extract isolation and Western blot

For Western blots, cells were lysed in radioimmunoprecipitation assay buffer (RIPA) (89901; Thermo Fisher Scientific) supplemented with protease inhibitor cocktail (P8340; Sigma-Aldrich) by adding 30 $\mu$l to each chamber and scratching cells with a scraper. Lysates from two chambers were pooled for each condition and mechanically burst using a 1-ml syringe and a 12-gauge needle. After centrifugation (10 min at 10,000$g$), the supernatant was

added to 4x Laemmli buffer (#1610747; Bio-Rad) with $\beta$-mercaptoethanol and boiled (5 min, 95°C). Proteins were separated using 4–20% SDS–PAGE gels (Bio-Rad) and blotted to nitrocellulose membranes (Sigma-Aldrich). Total protein staining was performed using Ponceau S solution (Sigma-Aldrich). Primary antibodies against LC3B (Map1lc3b) (Novus Biologicals), Bcar1 (BD Bioscience), paxillin (Pxn) (AHO0492; Invitrogen), talin (Tln1) (T3287; Sigma-Aldrich), vinculin (Vcl) (700062; Invitrogen), and Tomm20 (sc-17764; Santa Cruz) were detected by alkaline phosphatase–coupled secondary antibodies directed to the primary antibodies (A3562; A3812; A8438; Sigma-Aldrich). The total protein staining per lane was used as a loading control, and the normalized band intensities of stretched conditions were plotted as relative protein amount (percentage of the unstretched controls). A t test analysis was performed. The number of individual experiments is given in figure legends.

### Transcriptome analysis by qRT–PCR

RNA isolation was performed using the ROTIZOL RNA isolation kit (ROTH Art. No. 9319.1) following the manufacturer's protocol. Briefly, after stretching, cells were lysed directly in 100 $\mu$l of ROTH RNA solution. Cell lysis was achieved by gently pipetting the solution multiple times. The lysate was transferred to a microcentrifuge tube and incubated (10 min, RT). Subsequently, 20 $\mu$l chloroform was added, followed by vortexing before centrifugation (14,000$g$, 15 min, 4°C). The upper aqueous phase was transferred to a new tube, and RNA was precipitated with an equal volume of isopropanol. After incubation and centrifugation, the RNA pellet was washed with 75% ice-cold ethanol, air-dried, and resuspended in nuclease-free water.

cDNA synthesis was performed using QuantiTect Reverse Transcription Kit (QIAGEN). cDNAs of interest were quantified using 30 ng of total cDNA using StepOne Real-Time PCR System (Applied Biosciences, Thermo Fisher Scientific), TaqMan gene expression assay–specific primers (Hspa8: Rn00821191_g1; Pxn: Rn01499294_m1; Vcl: Rn01755886_m1; all with rat specificity and from Thermo Fisher Scientific), and master mix (Applied Biosciences, Thermo Fisher Scientific). Glyceraldehyde 3-phosphate dehydrogenase (GAPDH: Rn01462661_g1) was quantified as an endogenous control gene. For evaluation, StepOne Software (version 2.3) was used.

### ATP assay

For quantification of cellular adenosine triphosphate (ATP), CellTiter-Glo 2.0 Assay was used. For intracellular ATP levels, cells were washed with PBS, and 300 $\mu$l of 0.05% trypsin/EDTA solution was added to each chamber and incubated for 5 min to detach the adherent cells. To stop the reaction, 600 $\mu$l of growth medium was added and subsequently centrifuged (5 min, 220$g$) to remove the trypsin/EDTA and the pellet resuspended in growth medium. For measuring ATP levels, cell suspensions were first set to the same concentration of cells in 100 $\mu$l and combined with 100 $\mu$l of the CellTiter-Glo in a 96-well plate according to manufacturer's instructions (Promega). The contents were mixed for 2 min to ensure cell lysis and then incubated for 10 min to stabilize the luminescent signal. Luciferase catalyzes the mono-oxygenation of luciferin in the presence of magnesium, ATP, and molecular oxygen. The light produced is proportional to the concentration of ATP, and its intensity was measured in a plate reader (Tecan).

### RT2 Profiler PCR Array

The RT2 Profiler PCR Array contains gene-specific real-time/quantitative polymerase chain reaction (qRT-PCR) assays to analyze a set of 84 mitochondrial gene transcripts and three control elements. First, mature RNA was isolated using RNeasy Plus Mini Kit Cat. No./ID: 74136 in combination with the QIAshredder column Cat. No./ID: 79656, according to the manufacturer's instructions. RNA quality was determined using a spectrophotometer; RNA quality control parameters OD260/280 were between 1.8 and 2.0. Then, RNA was reverse-transcribed using the QIAGEN cDNA conversion kit. cDNA was used on real-time RT[2] Profiler PCR Array (Cat. no. PARN-087Z; QIAGEN) in combination with RT[2] SYBR Green qRT-PCR Master Mix (Cat. no. 330529). 25 $\mu$l (for 96-well plates) of the resulting mixture was loaded into each well of an RT2 Profiler PCR Array plate, which contained pre-loaded gene-specific primer sets for mitochondrial genes. Each array plate contained one set of 96 wells for testing. Genomic DNA contamination, reverse transcription, and positive PCR controls were included in each 96-well set on each plate. Cycle threshold (CT) values were exported to an Excel file to create a table of CT values. This table was then uploaded onto the data analysis web portal at http://www.qiagen.com/geneglobe. Samples were assigned to controls and test groups. CT values were normalized based on a Manual Selection of reference genes. The data analysis web portal calculates fold change/regulation using the delta CT method, in which delta CT is calculated between gene of interest (GOI) and an average of reference genes (HKG), followed by delta–delta CT calculations (delta CT [Test Group]–delta CT [Control Group]). Fold changes were then calculated according to the manufacturer's instructions and plotted in a heatmap as $\log_2$(FC) using the ComplexHeatmaps package (Gu et al, 2016).

### Python image analysis script availability

Custom Python image analysis scripts will be available upon request.

# Data Availability

Proteomics data have been deposited at the PRIDE repository with the accession number PXD060780.

# Supplementary Information

# Acknowledgements

The authors acknowledge the support from Jens Konrad in maintenance and manufacturing of stretching resources. This work was funded by the Deutsche Forschungsgemeinschaft (DFG, German Research Foundation) DFG-FOR 2743 (project number 388932620) to J Höhfeld, R Merkel, PF Huesgen, and B Hoffmann.

## Author Contributions

L Lövenich: conceptualization, data curation, formal analysis, validation, investigation, visualization, methodology, project administration, and writing—original draft, review, and editing.
B Rahimi: conceptualization, data curation, formal analysis, validation, investigation, visualization, methodology, project administration, and writing—original draft, review, and editing.
H Baeta: conceptualization, data curation, formal analysis, validation, investigation, visualization, methodology, project administration, and writing—original draft, review, and editing.
M Kuppusamy: investigation.
M Walkenbach: investigation.
F Rastfeld: investigation.
G Dreissen: software.
R Wein: software.
J Höhfeld: supervision, funding acquisition, and project administration.
R Merkel: supervision, funding acquisition, and project administration.
PF Huesgen: conceptualization, supervision, funding acquisition, project administration, and writing—review and editing.
B Hoffmann: conceptualization, supervision, funding acquisition, project administration, and writing—review and editing.

## Conflict of Interest Statement

The authors declare that they have no conflict of interest.

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
