## [Reviewer comments · Life Science Alliance]

Cyclic stretch induces autophagy-mediated focal adhesion remodeling and activates mitochondria

Lukas Lövenich, Bahareh Rahimi, Henrique Baeta, Maithreyan Kuppusamy, Moritz Walkenbach, Frederik Rastfeld, Georg Dreissen, Ronald Wein, Joerg Hoehfeld, Rudolf Merkel, Pitter Huesgen and Bernd Hoffmann

DOI: <https://doi.org/10.26508/lsa.202503347>

Corresponding author(s): Dr. Bernd Hoffmann (Forschungszentrum Jülich)

Review Timeline:

Submission Date:	2025-04-07
Editorial Decision:	2025-05-27
Revision Received:	2025-11-14
Editorial Decision:	2025-12-22
Revision Received:	2026-01-13
Accepted:	2026-01-19

Scientific Editor: Tim Fessenden

Transaction Report:

May 27, 2025

Re: Life Science Alliance manuscript #LSA-2025-03347-T

Dr. Bernd Hoffmann
Forschungszentrum Jülich
Forschungszentrum Jülich GmbH/ICS-7 Biomechanics
Research Centre Juelich, Institute of Bio- and Nanosystems, IBN-4
Jülich 52425
Germany

Dear Dr. Hoffmann,

Thank you for submitting your manuscript entitled "Cyclic stretch induces autophagy-mediated focal adhesion remodeling and activates mitochondria" to Life Science Alliance. The manuscript was assessed by expert reviewers, whose comments are appended to this letter.

As you will see, reviewers appreciated the novel connections established between cytoskeletal remodeling and autophagy with additional new observations on mitochondrial biomass and function. All reviewers remarked on key areas in which improved evidence is needed to support conclusions, as well as needed methodology details and discussion points. Namely, all reviewers noted the mitochondrial dye used does not report on membrane potential, and an appropriate probe should be used to support this claim. Reviewers 1 and 3 both requested improved measures of autophagic flux either with a western blot time course or with tandem fluorescent reporters. The important discrepancies noted by Reviewer 1 in points 3 and 4 should either be discussed or resolved with new data. Finally Reviewers 2 and 3 made several important suggestions to clarify data presentation and methods. While we concur that the precise role of mitochondrial biomass and energetics is not causally linked with the rest of the manuscript, the nature of this connection can be discussed and no new data are required here.

Thank you for this interesting contribution to Life Science Alliance. We are looking forward to receiving your revised manuscript.

Sincerely,

B. MANUSCRIPT ORGANIZATION AND FORMATTING:

Reviewer #1 (Comments to the Authors (Required)):

Lövenich et al. investigate the mechanistic adaptation of smooth muscle cells to uniaxial cyclic stretch. The authors demonstrate that mechanical stress induces a Bag3-dependent autophagy (CASA) response that remodels focal adhesions and promotes actin cytoskeletal reorientation. They further show data that stretch increases the mitochondrial mass and results in an accumulation of mitochondrial proteins and transcripts, associated with this cytoskeletal mechanoresponse. The study presents a conceptually novel link between mechanical force, CASA-mediated proteostasis, and mitochondrial function.

The study presents a comprehensive and timely contribution to the field, specifically a novel link between cell mechanics, autophagy and cytoskeletal regulation of cell dynamics.

Methodologically, the work is rigorous and technically well-executed, as it integrates live-cell stretching, pharmacological and genetic inhibition of autophagy, phosphomutant analysis, and label-free quantitative proteomics. These approaches are thoughtfully interconnected to demonstrate structural and metabolic responses to mechanical strain. The choice of a physiologically relevant model (A7r5 smooth muscle cells) strengthens the relevance of the findings, which are broadly applicable to vascular biology and mechanotransduction research.

As moderate shortcomings, direct evidence for sustained or productive autophagic flux is lacking and the data on mitochondrial activation are technically flawed. The data on autophagy degrading focal adhesion components and thereby directly affecting cell dynamics are so far inconclusive.

1. The claim that cyclic stretch "induces autophagy, characterized by a rapid formation of autophagosomes and their turnover after 4 h" is not supported by direct evidence. While imaging suggests a peak in LC3B+ structures at 10 min and a decline by 4 h, this does not directly prove productive autophagic flux. More direct flux assays (e.g., p62, LC3-II time course Wb, or tandem fluorescent reporters) are necessary to support the functional engagement of autophagy pathways. A time-course of LC3B-I/II levels, ideally with and without CQ, and inclusion of flux markers such as p62/SQSTM1 are required to support this key claim.

2. Proteomics analysis shows that the CASA components (Bag3, Hspa8, LC3B) were depleted at 4 h post-stretch, yet, puzzlingly, the authors claim that CASA is actively involved in cytoskeletal remodeling until this stage. Depletion could reflect exhaustion or transcriptional downregulation. qPCR data or time-resolved analysis at protein level would help resolve this.

3. The authors claim that autophagy depletes focal adhesion proteins and thereby affects their function, however the Western blots cited as validating proteomics results only show significant downregulation for paxillin. Other tested FA proteins (Vcl, Bcar1, Tln1, Flna) do not show significant depletion or restoration upon autophagy inhibition. This weakens the claim that these are degraded via CASA. Thus, it is not resolved whether direct or indirect, regulation effects were observed. The authors should reconsider their interpretation or provide stronger evidence for either option.

4. The claim that chloroquine rescues focal adhesion protein levels and supports their degradation via autophagy is not convincing. Upon validation of the proteomic data after 4 h of stretch, only paxillin exhibited a statistically significant reduction. The other tested proteins (vinculin, p130Cas, talin, and filamin A) show only non-significant trends, and none are restored by chloroquine treatment. This weakens the claim that these proteins are bona fide targets of autophagy under stretch conditions.

The interpretation should be supported by additional evidence or modified.

5. MitoTracker Green FM was used to infer increased mitochondrial activity, yet this dye is not sensitive to membrane potential. It reflects mitochondrial mass. Thus, the authors cannot conclude that mitochondrial activity is increased based on this data alone. Use of potential-sensitive dyes (e.g., TMRE, JC-1) or Seahorse analysis will be required to support their claim.

Minor points

6. The Abstract should mention that A7r5 is a rat smooth muscle cell line.

7. Figure 1C: Add arrows to indicate stretch direction.

8. Figure S1 legend: Actin is described as red but appears white in the image.

9. Figure 2A: The reported 1.4 AP/cell is not visually apparent in the image. Consider adding arrowheads.

10. Figure S4: Add condition labels (e.g., unstr., 4h str., 4h str. + CQ) to WB plots.

11. Chloroquine concentration and treatment duration are missing in the Methods section.

12. Statistical tests (e.g., K-S tests, *p-values) should be specified in all figure legends. Likewise, the number of independent biological experiments needs to be indicated in all figure legends.

13. Custom Python image analysis scripts should be made available (e.g., via GitHub) or upon request.

Reviewer #2 (Comments to the Authors (Required)):

This study provides an analysis of the cellular responses of A7r5 rat smooth muscle cells to mechanical stretch. The data demonstrates that stretching induces adaptive actin fibre reorientation and triggers autophagy, which plays a crucial role in subsequent cellular remodelling. The authors implicate chaperone-assisted selective autophagy through Bag3 in particular. Proteomic analysis reveals that mechanical strain leads to the depletion of cytoskeletal and focal adhesion proteins but inhibiting autophagy can reduce this depletion. The study also shows that stretching causes a reduction in focal adhesion size, with the remaining focal adhesions reorienting perpendicularly to the direction of strain. The authors also indicate that prolonged stretching activates mitochondria, and inhibiting ATP synthase can impair this actin reorganisation. This is an interesting investigation that suggests a connection between mechanical stretch and autophagy, as well as mitochondria metabolism. Some of the experiments and presentation of figures could be improved, as suggested below, to make the study more convincing.

Comments and suggestions for revisions:

1. It is not clear what the analysis of single cells vs confluent cells adds. The so-called single cells often seem to be touching other cells in the images- e.g. Figure 1A and B. Also, there seems to be no difference. Maybe this part can be moved to supplementary or removed. Maybe instead of single cells, they could be referred to as subconfluent cells?
2. MitoTracker Green FM accumulation as an indication of increased membrane potential over time is potentially unreliable. It would be better suited to use ratiometric dyes like TMRE/TMRM/JC-1 if possible. MTGreen expression is not strictly membrane potential-dependent as with other MitoTracker probes and is unable to leave mitochondria after uptake, so its accumulation over time in mitochondria isn't the strongest method for assessing membrane potential.
3. Stretch direction should be labelled on images - orientation angles and alignment of fibres with stretch were not clear. Use of parallel and perpendicular to stretch is unclear, looks like fibres align with stretch direction which would be parallel to stretch direction rather than perpendicular.
4. It isn't clear always whether the authors have performed biologically independent repeats on different days or just technical replicates. This should be specified and it is preferable to perform independent biological replicates.

Reviewer #3 (Comments to the Authors (Required)):

Review of "Cyclic stretch induces autophagy-mediated focal adhesion remodeling and activates mitochondria" by Lövenich et al. Summary: In this manuscript, Lövenich et al. investigate how cyclic mechanical stretch affects cytoskeletal organization and mitochondrial activity in A7r5 smooth muscle cells. They compare cells grown as isolated single cells versus monolayers, showing that both exhibit actin fiber reorientation perpendicular to strain. They report that uniaxial stretch rapidly induces autophagy (quantified by LC3 puncta and LC3-I/II conversion) and that inhibition of autophagy (via chloroquine or a CASA-deficient Bag3 mutant) slows actin reorientation. Proteomic analysis after 4 h stretch ({plus minus} chloroquine) reveals depletion

of focal adhesion proteins and CASA regulators, and an enrichment of mitochondrial proteins. Consistent with this, Western blots and immunostaining show decreased levels and size of focal adhesion components (paxillin, vinculin, etc.) under stretch. The authors further report that mitochondrial content (Tomm20 area/number) is unchanged, but staining with MitoTracker Green FM increases after 4 h stretch. They interpret this as increased mitochondrial membrane potential, although cellular ATP levels remain unchanged. Finally, inhibiting ATP synthase with oligomycin similarly slows stretch-induced actin reorientation. The authors conclude that cyclic stretch triggers CASA-dependent autophagy to remodel focal adhesions and activates mitochondrial function to support the adaptive mechanoresponse.

Novelty and Contribution: The study addresses an important question in mechanobiology: how cells adapt to sustained mechanical stress. Linking cyclic stretch to autophagy-mediated turnover of focal adhesions is novel and supported by a multi-pronged approach (imaging, biochemistry, proteomics). The finding that mitochondria are "activated" by stretch is intriguing and suggests a coordination between energy metabolism and cytoskeletal adaptation. Overall, the work advances our understanding of how mechanotransduction intersects with autophagy and mitochondrial function. The combination of proteomics and functional assays is a strength. However, some conclusions (particularly regarding mitochondrial activation) are not fully substantiated by the data presented.

Methodology and Data Quality: The experimental approaches are generally appropriate. The authors use well-established assays for autophagy (LC3 imaging and flux analysis with chloroquine) and for focal adhesion analysis (immunostaining for paxillin and vinculin). The cyclic stretch protocol (duration, strain amplitude, frequency) could have been more clearly described in the Methods. It is positive that the authors included both immunofluorescence and quantitative proteomics, but some methodological details need clarification:

- **Autophagy Assay:** The quantification of autophagic puncta ("APs") is convincing, and the use of chloroquine to trap LC3-II is appropriate. It would help to state the imaging and analysis criteria for LC3 puncta (e.g. size threshold, normalization to cell area). The decrease in LC3-II without chloroquine after 4 h and its accumulation with chloroquine is consistent with increased flux, but it is not fully demonstrated whether this represents increased autophagy induction or simply turnover. A time course of LC3-II/I with and without lysosomal blockade at earlier times (e.g. 10 min, 1 h) would strengthen the conclusion of early autophagy induction.
- **Cytoskeletal Reorientation:** The actin fiber orientation analysis seems robust, and the use of confidence intervals is helpful. However, it would clarify interpretation if the authors stated the number of cells analyzed per condition and whether multiple independent experiments were performed.
- **Mitochondrial Assays:** The authors assess mitochondria by imaging (Tomm20, MitoTracker Green) and by measuring cellular ATP. While Tomm20 area/number shows no change, the MitoTracker staining increased. Importantly, MitoTracker Green FM is generally considered to label mitochondria independently of membrane potential, so its increased intensity is not a definitive indicator of higher potential. The conclusion that membrane potential increased is therefore speculative based on these data. Moreover, ATP levels did not change, raising questions about the functional significance. Direct measurement of membrane potential (e.g. with JC-1) would be a more appropriate assay. Alternatively, an assay of respiration (OCR) or other functional readout could help. The authors do use oligomycin (ATP synthase inhibitor) to show that ATP production contributes to actin reorientation, which supports a metabolic role. However, the link between stretch and mitochondrial "activation" requires more evidence.

Clarity and Figure Presentation: The manuscript is generally well organized and the figures are clear. The labeling in figures (axes, units, scale bars) should be checked carefully. For example, some fluorescence images (e.g. in Figure 1) do not have arrowheads to illustrate stretch direction. In the text, it would be clearer to define term abbreviations at first use (e.g. "FA" for focal adhesion in the abstract). There are a few language issues (e.g. "did not observed differences" should read "did not observe differences" in Results line 269). The narrative generally flows well from autophagy to focal adhesions to mitochondria, although the jump to proteomics and mitochondria is a bit abrupt; perhaps a bridging sentence emphasizing the link to metabolic adaptation would help. Figures 2 and 6 (showing quantitative analyses) are generally well presented. It would be helpful if all quantitative plots included "the number of replicates" in addition to cells analyzed (n) in the legend.

Major Comments:

1. **Mitochondrial activity assessment (Figure 7D-E):** The reported increase in MitoTracker Green FM staining after 4 h of stretch is intriguing, but the imaging needs careful control. In Figure 7D, the background fluorescence in the stretched vs unstretched images appears different, which could artificially influence the intensity measurement. I strongly recommend showing representative images side-by-side with identical microscope settings (same exposure, gain, and background correction) for the unstretched and stretched conditions. Moreover, include quantification of background-corrected fluorescence intensity (e.g. mean intensity per cell after subtracting background) and indicate the number of cells analyzed. This will ensure that the observed increase is not an artifact of imaging conditions.
2. **Interpretation of MitoTracker signal and suggestion for direct membrane potential measurement:** The authors interpret the increased MitoTracker Green FM signal as evidence of higher mitochondrial membrane potential. However, MitoTracker Green is known to accumulate in mitochondria largely independent of membrane potential; it is often used as a mitochondrial mass marker rather than a potential indicator. The lack of change in mitochondrial number/area (Figure 7A-C) suggests that the signal change could be due to altered dye uptake or mitochondrial membrane composition, not necessarily potential. To robustly test mitochondrial membrane potential, I suggest performing an experiment using a well-established potentiometric dye, such as JC-1. These dyes respond to changes in membrane potential and would directly address whether stretching increases mitochondrial polarization. If new experiments are not feasible before publication, the authors could rephrase conclusions to reflect that MitoTracker intensity changes are correlative and suggest potential changes, rather than definitive proof. At minimum, the authors should acknowledge this limitation more explicitly in the Discussion and moderate the claim that mitochondrial potential is increased.

3. Autophagy flux and temporal dynamics: The data convincingly show a spike in autophagic puncta at 10 min of stretch and normalization by 4 h, with chloroquine causing sustained puncta. While the LC3-I/II blots support increased turnover, the interpretation would be stronger with a clearer depiction of flux. For example, showing LC3 blots (plus/minus chloroquine) at the 10-min time point could confirm that the early autophagy spike is real. The current data focus on 4 h with/without CQ. The authors should also clarify how "APs" were counted in imaging (e.g. criteria for a punctum) and note whether any differences in cell morphology under CQ could affect puncta counts. Overall, explicitly stating that stretch induces a transient burst of autophagy (peaking at ~10 min) would help clarify the dynamic nature of this response.

4. Mechanistic clarity and CASA involvement: The manuscript emphasizes the role of Bag3-mediated CASA in this process. The use of a phospho-mimetic Bag3 mutant is a strength. However, it would help to clearly delineate which observations are specific to CASA. For instance, the proteomics shows depletion of CASA regulators (Bag3, Hsp8, etc.) after stretch. Are any of the accumulated mitochondrial proteins known Bag3 clients, or is there a hypothesized link between CASA and mitochondrial regulation? The authors might consider a brief discussion of whether CASA (or chaperone-mediated pathways) could directly influence mitochondrial quality control or biogenesis under mechanical stress. As is, the link is more correlative. Emphasizing the distinction between CASA's effect on cytoskeletal proteins and the separate effect on mitochondria would improve mechanistic clarity.

5. Introduction - discuss cytoskeletal control of ATP. The Introduction frames a link between cytoskeleton and metabolism, but it would benefit from citing very recent work on this topic. Several studies have established that mechanical or cytoskeletal cues directly regulate glycolytic flux and ATP production. For example, Hu et al. showed that PI3K-mediated actin remodeling releases aldolase A from F-actin, leading to a boost in glycolysis (PMID: 26824656). Similarly, Park et al. demonstrated that substrate stiffness mechanically controls glycolysis: softening the matrix caused stress fiber disassembly and degradation of phosphofructokinase, thereby downregulating glycolysis (PMID: 32051585). More recently, Zhan et al. (PMID: 38328193) described dynamic cortical "glycolytic waves" driven by Ras/PI3K-F-actin networks, which locally supply ATP for protrusion and migration. These findings underscore that cytoskeletal architecture and tension influence ATP levels and metabolic enzymes. The authors are encouraged to briefly discuss these studies in the Introduction (or Discussion) to place their mitochondria and ATP results in context.

Minor Comments:

- In the Abbreviations section, "AP, autophagosome" and "ATP, adenosine triphosphate" appear on the same line, which is confusing. Please separate them clearly. Also define "APs" as autophagosomes at first use in the main text.
- On p.2 (lines 267-269), correct the grammar: "we did not observed differences" should be "we did not observe differences".
- In the Methods, please specify the parameters of the cyclic stretch (amplitude, frequency, duty cycle). This is important for reproducibility.
- In Figure legends, always state the number of biological replicates (n) or cells analyzed. For example, in Figure 2 and 6, indicate how many cells or wells contributed to the quantified data.
- In Figure 1 and other orientation plots, clarify how the angle distribution was computed (e.g. direction of stretch = 0{degree sign}, parallel = 0-5{degree sign} = "parallel", etc.). Also, mention the total number of cells analyzed per condition.
- The discussion could briefly mention other studies linking mechanotransduction and mitochondrial function (e.g. via calcium signaling or ER stress) to contextualize the findings.
- Ensure statistical tests and p-values are described in the Methods figure legends. Some figures label "" for p<0.05; please define these and mention the test used in the legends.

Recommendation: The manuscript presents an interesting study at the intersection of mechanobiology, autophagy, and mitochondrial biology. The experimental work is largely sound and the multi-pronged approach is comprehensive. However, the interpretation of mitochondrial activation needs strengthening, and some quantitative clarifications (as noted above) are required. I recommend major revision. Addressing the key points about mitochondrial assays (adding or revising data on membrane potential, ensuring imaging controls for Figure 7) and clarifying the autophagy-focal adhesion link will greatly improve the rigor and clarity of the study. Upon revision, the manuscript will make a valuable contribution to understanding how cells adapt to mechanical stretch.

Dear Tim Fessenden,

Scientific Editor of the Journal of Life Science Alliance,

We are resubmitting our revised manuscript entitled 'Cyclic stretch induces autophagy-mediated focal adhesion remodeling and activates mitochondria' by Lövenich, Rahimi, Baeta et al. (Manuscript ID: LSA-2025-03347-T), which we have revised according to the reviewers' and editor's comments. We appreciate the constructive feedback and believe that the revisions have substantially improved the manuscript.

As suggested by all reviewers, we have extended our mitochondria data set using a dye that reports on the membrane potential. Further, we have improved the measurement of the autophagic flux by western blot time course as indicated by Reviewer 1 and 3.

We took concerns by Reviewer 1 (points 13 and 4) regarding the discrepancies between mass spectrometry results and western blot data on protein degradation very seriously. We repeated the western blot experiments with optimized lysis and conditions which gave more consistent results. We further weakened the statement of general FA protein degradation upon stretching and focused more on paxillin and clearly stated that although paxillin is degraded by autophagy, our results do not establish a clear link to the CASA machinery. Further research must be examined here to clarify the exact degradation mechanism.

Moreover, we appreciated the suggestions of Reviewer 2 and 3 to clarify data presentation and methods to improve our manuscript. We further discussed the connection between stress responses, mitochondria and cellular bioenergetics in greater depth.

Detailed responses to all reviewers' and editors' comments are appended below in red colored text.

We sincerely thank the reviewers and the editor for their helpful suggestions and valuable comments, which have significantly enhanced the quality of our work. We hope that the revised version now meets the expectations and are looking forward to your favorable consideration of our manuscript.

Yours sincerely

Bernd Hoffmann

Reviewer comments Lövenich et al.

Manuscript #	LSA-2025-03347-T
Title	Cyclic stretch induces autophagy-mediated focal adhesion remodeling and activates mitochondria
Corresponding Author	Dr. Bernd Hoffmann (Forschungszentrum Jülich)
Date:	2025-05-27 08:37:41
Last Sent:	2025-05-27 08:37:41
Created By:	Redacted
From:	t.fessenden@life-science-alliance.org
To:	b.hoffmann@fz-juelich.de
BCC:	Redacted
Subject:	Life Science Alliance Manuscript - Editorial Decision LSA-2025-03347-T
Email	May 27, 2025 Re: Life Science Alliance manuscript #LSA-2025-03347-T Dr. Bernd Hoffmann Forschungszentrum Jülich Forschungszentrum Jülich GmbH/ICS-7 Biomechanics Research Centre Juelich, Institute of Bio- and Nanosystems, IBN-4 Jülich 52425 Germany Dear Dr. Hoffmann, Thank you for submitting your manuscript entitled "Cyclic stretch induces autophagy-mediated focal adhesion remodeling and activates mitochondria" to Life Science Alliance. The manuscript

was assessed by expert reviewers, whose comments are appended to this letter.

As you will see, reviewers appreciated the novel connections established between cytoskeletal remodeling and autophagy with additional new observations on mitochondrial biomass and function. All reviewers remarked on key areas in which improved evidence is needed to support conclusions, as well as needed methodology details and discussion points. Namely, all reviewers noted the mitochondrial dye used does not report on membrane potential, and an appropriate probe should be used to support this claim. Reviewers 1 and 3 both requested improved measures of autophagic flux either with a western blot time course or with tandem fluorescent reporters. The important discrepancies noted by Reviewer 1 in points 3 and 4 should either be discussed or resolved with new data. Finally Reviewers 2 and 3 made several important suggestions to clarify data presentation and methods. While we concur that the precise role of mitochondrial biomass and energetics is not causally linked with the rest of the manuscript, the nature of this connection can be discussed and no new data are required here.

The typical timeframe for revisions is three months. Please note that papers are generally considered through only one revision cycle, so strong support from the referees on the revised version is

needed for acceptance.

Thank you for this interesting contribution to Life Science Alliance. We are looking forward to receiving your revised manuscript.

Sincerely,

- A letter addressing the reviewers' comments point by point.
- An editable version of the final text (.DOC or .DOCX) is needed for copyediting (no PDFs).
- High-resolution figure, supplementary figure and video files uploaded as individual files: See our detailed guidelines for preparing your production-ready images, <https://www.life-science-alliance.org/authors>
- Summary blurb (enter in submission system): A short text summarizing in a single sentence the study (max. 200 characters including spaces). This text is used in conjunction with the titles of papers, hence should be informative and complementary to the title and running title. It should describe the context and

significance of the findings for a general readership; it should be written in the present tense and refer to the work in the third person. Author names should not be mentioned.

B. MANUSCRIPT ORGANIZATION AND FORMATTING:

Reviewer #1 (Comments to the Authors (Required)):

Lövenich et al. investigate the mechanistic adaptation of smooth muscle cells to uniaxial cyclic stretch. The authors demonstrate that mechanical stress induces a Bag3-dependent autophagy (CASA) response that remodels focal adhesions and promotes actin cytoskeletal reorientation. They further show data that stretch increases the mitochondrial mass and results in an accumulation of

mitochondrial proteins and transcripts, associated with this cytoskeletal mechanoresponse. The study presents a conceptually novel link between mechanical force, CASA-mediated proteostasis, and mitochondrial function.

The study presents a comprehensive and timely contribution to the field, specifically a novel link between cell mechanics, autophagy and cytoskeletal regulation of cell dynamics.

Methodologically, the work is rigorous and technically well-executed, as it integrates live-cell stretching, pharmacological and genetic inhibition of autophagy, phosphomutant analysis, and label-free quantitative proteomics. These approaches are thoughtfully interconnected to demonstrate structural and metabolic responses to mechanical strain. The choice of a physiologically relevant model (A7r5 smooth muscle cells) strengthens the relevance of the findings, which are broadly applicable to vascular biology and mechanotransduction research.

As moderate shortcomings, direct evidence for sustained or productive autophagic flux is lacking and the data on mitochondrial activation are technically flawed. The data on autophagy degrading focal adhesion components and thereby directly affecting cell dynamics are so far inconclusive.

1. The claim that cyclic stretch "induces autophagy, characterized by a rapid formation of autophagosomes and their turnover after 4 h" is not supported by direct evidence. While imaging suggests a peak in LC3B+ structures at 10 min and a decline by 4 h, this does not directly prove productive autophagic flux. More direct flux assays (e.g., p62, LC3-II time course Wb, or tandem fluorescent reporters) are necessary to support the functional engagement of autophagy pathways. A time-course of LC3B-I/II levels, ideally with and without CQ, and inclusion of flux markers such as p62/SQSTM1 are required to support this key claim.

We understand the reviewer's request for experimental data on

autophagic flux. Unfortunately, we could not blot for a second flux reporter because the tested anti-sqstm1/p62 antibody failed to detect the protein.

As suggested, we analyzed the LC3B-I and -II time course across the stretching timeframe (10 min, 30 min, 1 h and 4 h compared to unstretched controls). We used CQ treatment in both unstr. and 4 h str. conditions as a control for impaired flux. Stretching lead to a quick increase of LC3B-II relative to LC3B-I, which returned after 4 h of stretch. CQ treatment led to accumulation of the lipidated (membrane form) of LC3B, the lower band of LC3B-II, consistent with the impairment of AP turnover. As before, we consistently observed that the levels of both LC3B-II and LC3B-I were lower after 4 h of stretching than to the earlier stretching time points (30 min and 1 h). Taken together, in our opinion these results add to the microscopy and the proteomics data, arguing for a functional engagement of autophagy upon stretching, where the autophagic flux, measured by autophagosome formation (higher lc3b-II/lc3b-I ratio) and consumption (lower ratio and lower levels of both lc3b-II and lc3b-I) are noticeable. The new data has been added to Figure 2 as panels C and D.

The previous blots and quantitative analysis for 4h str. +/-CQ have moved to Figure S2 as panels C and D. In addition, we have analyzed changes in LC3B-I and LC3B-II abundance after 10 min of stretching + and – CQ treatment. Here, we could observe a tendency increased LC3B-II abundance after stretch, but did not reach a significant threshold, most probably due to the low number of repeats (n=3). The data has been added to supplementary Figure S2 A and B to support the overall data set.

2. Proteomics analysis shows that the CASA components (Bag3, Hspa8, LC3B) were depleted at 4 h post-stretch, yet, puzzlingly, the authors claim that CASA is actively involved in cytoskeletal remodeling until this stage. Depletion could reflect exhaustion or transcriptional downregulation. qPCR data or time-resolved analysis at protein level would help resolve this.

The reviewer is right, that decreased abundance of the CASA components or also the other proteins depleted after 4 h of stretch could be due to exhaustion or transcriptional downregulation. For that reason, we have performed qPCR analysis to quantify their RNA levels to exclude transcriptional downregulation. Here we have focused on the FA proteins paxillin and vinculin and the CASA component Hspa8, which were depleted after 4 h of stretching both in mass spectrometry and western blot analysis.

Here, we could nicely observe that the mRNA level of paxillin was significantly increased, while the mRNA levels of vinculin and Hsc70/Hspa8 remained unchanged after 1h or 4h of uniaxial cyclic stretching compared to the unstretched control. With that we show that neither the FA proteins paxillin or vinculin, nor the Hsc70 are depleted during uniaxial cyclic stretching due to transcriptional downregulation. In addition, the increase in paxillin mRNA levels may indicate a compensatory response to stretch-induced degradation. This is in agreement with initial descriptions made (Ulbricht et al 2013), where cells dispose damaged filamins via CASA, while increasing their transcription, to produce new filamin and effectively remodel their mechanosensitive structures.

3. The authors claim that autophagy depletes focal adhesion proteins and thereby affects their function, however the Western blots cited as validating proteomics results only show significant downregulation for paxillin. Other tested FA proteins (Vcl, Bcar1, Tln1, Flna) do not show significant depletion or restoration upon autophagy inhibition. This weakens the claim that these are degraded via CASA. Thus, it is not resolved whether direct or indirect, regulation effects were observed. The authors should reconsider their interpretation or provide stronger evidence for either option.

We agree with the reviewer that our statements were too generalized. Therefore, we have adapted several points in our manuscript.

First, we have repeated the WB after 4h of stretching with and

without CQ treatment to enhance the impact of the data set. Here, we focused on the proteins paxillin and vinculin as further analysis focuses on these FA proteins. Additionally, we analyzed the chaperone Hsc70/Hspa8 as a component of the CASA complex.

We could see that paxillin is not just significantly depleted after 4h of stretching, but that this could be prevented when using CQ treatment in parallel to stretching (Figure S4B). Secondly, for Hsc70/Hspa8 we observed an identical trend as shown for paxillin, indicating to be a potential client of CASA (depletion after stretching and block when using CQ treatment while stretching). Unfortunately, we could not repeat the WBs for the other FA proteins (Filamin A, Talin and p130Cas). Since these FA proteins were not followed up in any further analysis (PCR, immunostainings), we decided to remove these blots from the manuscript to maintain the logical flow, clarity, and consistency. We now focus on paxillin rather than general FA protein abundance which shows a stronger effect than indicated in the initial submitted data set.

We agree that our data does not provide a direct link to CASA. We have reconsidered the interpretations made in our manuscript as suggested and discussed a more general autophagy-mediated degradation of paxillin and the potential CASA connection, which requires future investigation.

4. The claim that chloroquine rescues focal adhesion protein levels and supports their degradation via autophagy is not convincing. Upon validation of the proteomic data after 4 h of stretch, only paxillin exhibited a statistically significant reduction. The other tested proteins (vinculin, p130Cas, talin, and filamin A) show only non-significant trends, and none are restored by chloroquine treatment. This weakens the claim that these proteins are bona fide targets of autophagy under stretch conditions. The interpretation should be supported by additional evidence or modified.

We agree with the statement of the reviewer and as discussed in point 3, we have repeated the western blots and could see that for

paxillin the degradation after stretching could be blocked when using CQ treatment providing additional evidence. This indicates that autophagy is involved in the stretch induced depletion of paxillin. We agree that our data does not show that the degradation is mediated via CASA or any particular autophagy pathway. We have made that point more careful in our discussion referring that it needs further examination.

As mentioned above, Western blot data for the other proteins analyzed (vinculin, p130cas, talin and filamin A) could not be repeated and for logical flow and consistency of the manuscript we have decided to remove them. The blots for paxillin, vinculin and Hsc70 remain within our manuscript as we continued with paxillin and vinculin in further analysis and HSC70 as CASA complex component shows and identical trend as paxillin.

We hope that our data and the corrections in the discussion are now convincing in the reviewers' eyes and useful for the readers' interpretation of our manuscript.

5. MitoTracker Green FM was used to infer increased mitochondrial activity, yet this dye is not sensitive to membrane potential. It reflects mitochondrial mass. Thus, the authors cannot conclude that mitochondrial activity is increased based on this data alone. Use of potential-sensitive dyes (e.g., TMRE, JC-1) or Seahorse analysis will be required to support their claim.

We agree with the reviewer's interpretation that Mitotracker staining cannot be correlated with membrane potential and is more reflective of mitochondrial mass. Therefore, we have moved the MitoTracker as well as the Tomm20 data to the Supplementary Figure S6D-H.

As suggested, we probed the mitochondrial membrane potential in A7r5 ML following stretching using the JC-1 assay. We observed increased mitochondrial membrane potential after 4 h of stretching as indicated by an increased red fluorescent signal. We believe that these results strengthen the interpretation that mitochondria were activated following stretching. This supports the hypothesis,

in addition to the increased mitochondrial protein and transcript levels, and the imaging data that mitochondria were activated in response to cyclic stretching. We therefore adjusted Figure 7A-C to include the JC-1 results, more relevant for the readers, specifically when presented along with the effect of oligomycin inhibition on cytoskeletal orientation, revealing that the mitochondrial activity (namely ATP synthase function) contributes to the cellular mechanoresponse.

Minor points

6. The Abstract should mention that A7r5 is a rat smooth muscle cell line.

We have added that A7r5 is a rat smooth muscle cell line to the Abstract.

7. Figure 1C: Add arrows to indicate stretch direction.

We have added arrows indicating stretch direction for better visualization to Figure 1C.

8. Figure S1 legend: Actin is described as red but appears white in the image.

Well recognized. We have changed the Figure legend description to 'white' for actin.

9. Figure 2A: The reported 1.4 AP/cell is not visually apparent in the image. Consider adding arrowheads.

We clearly understand that point and have therefore added red arrowheads indicating the 1.4 APs/cell in Figure 2A for the unstretched control group making the visual appearance clearer.

10. Figure S4: Add condition labels (e.g., unstr., 4h str., 4h str. + CQ) to WB plots.

Absolutely correct. We have added the condition labels unstr., 4 h str. – and 4 h str. +CQ below the Western Blot data as suggested.

11. Chloroquine concentration and treatment duration are missing in the Methods section.

Chloroquine concentration and treatment duration have been added to the Methods section 'LC3B-spots analysis' for better traceability.

12. Statistical tests (e.g., K-S tests, *p-values) should be specified in all figure legends. Likewise, the number of independent biological experiments needs to be indicated in all figure legends.

As requested, we have added statistical tests specifically for all analysis as well as the number of independent biological experiments to all Figures legends and Supplementary Figure legends.

13. Custom Python image analysis scripts should be made available (e.g., via GitHub) or upon request.

We have added a section for 'Python image analysis scripts availability' at the end of the Methods part to clarify Python script availability upon request.

Reviewer #2 (Comments to the Authors (Required)):

This study provides an analysis of the cellular responses of A7r5 rat smooth muscle cells to mechanical stretch. The data demonstrates that stretching induces adaptive actin fibre reorientation and triggers autophagy, which plays a crucial role in subsequent cellular remodelling. The authors implicate chaperone-assisted selective autophagy through Bag3 in particular. Proteomic analysis reveals that mechanical strain leads to the depletion of cytoskeletal and focal adhesion proteins but inhibiting autophagy can reduce this depletion. The study also shows that stretching causes a reduction in focal adhesion size, with the remaining focal adhesions reorienting perpendicularly to the direction of strain. The authors also indicate that prolonged stretching activates mitochondria, and inhibiting ATP synthase can impair this actin reorganisation. This is an interesting investigation that suggests a connection between mechanical stretch and autophagy, as well as mitochondria metabolism. Some of the experiments and presentation of figures could be improved, as suggested below, to make the study more convincing.

Comments and suggestions for revisions:

1. It is not clear what the analysis of single cells vs confluent cells adds. The so-called single cells often seem to be touching other cells in the images- e.g. Figure 1A and B. Also, there seems to be no difference. Maybe this part can be moved to supplementary or removed. Maybe instead of single cells, they could be referred to as subconfluent cells?

We understand that comment and have changed SC as abbreviation to subconfluent (SC cells = subconfluent cells). In Figure 1 A-D and Supp. Figure 1A we have changed the legend to SC cells and ML cells as well as in the Figure legends and the main text.

We understand the idea to remove Figure 1A and B to the supplement but decided to keep it as it is to emphasize the connection to prior data on subconfluent cells. If the reviewer would like to remove it, we would of course do so.

2. MitoTracker Green FM accumulation as an indication of increased membrane potential over time is potentially unreliable. It would be better suited to use ratiometric dyes like TMRE/TMRM/JC-1 if possible. MTGreen expression is not strictly membrane potential-dependent as with other MitoTracker probes and is unable to leave mitochondria after uptake, so its accumulation over time in mitochondria isn't the strongest method for assessing membrane potential.

We agree with the reviewer's interpretation that Mitotracker staining cannot be correlated with membrane potential. Therefore, we have removed the MitoTracker staining, as well as the Tomm20 data to the Supplement and performed analysis of JC-1 as recommended by the reviewer. Here, we observed increased mitochondrial membrane potential after 4 h of stretching. We believe these results strengthen the interpretation that mitochondria were activated following stretching. This adds to the increase in mitochondrial protein and transcript levels, and the imaging data presented before. We therefore adjusted Figure 7 to include the JC-1 results, more relevant for the readers, specifically when presented along with the effect of oligomycin inhibition on cytoskeletal orientation. We kept the Mitotracker and Tomm20 in the supplement (Figure S6D-H).

3. Stretch direction should be labelled on images - orientation angles and alignment of fibres with stretch were not clear. Use of parallel and perpendicular to stretch is unclear, looks like fibres align with stretch direction which would be parallel to stretch direction rather than perpendicular.

Well recognized. We have added the arrowheads for indication of stretch direction to Figure 1 C, where it was missing. Now it should be clearer that actin fibers orient due to stretching perpendicular to the stretch direction in all our experiments.

4. It isn't clear always whether the authors have performed

biologically independent repeats on different days or just technical replicates. This should be specified and it is preferable to perform independent biological replicates.

We totally agree to the reviewers' comments that biologically independent repeats are preferred rather than just technical replicates. Therefore, we have added the number of independently performed biological replicates (on different days/weeks!) and if applicable the number of technical replicates to all figure legends and supplementary figure legends.

Reviewer #3 (Comments to the Authors (Required)):

Review of "Cyclic stretch induces autophagy-mediated focal adhesion remodeling and activates mitochondria" by Lövenich et al.

Summary: In this manuscript, Lövenich et al. investigate how cyclic mechanical stretch affects cytoskeletal organization and mitochondrial activity in A7r5 smooth muscle cells. They compare cells grown as isolated single cells versus monolayers, showing that both exhibit actin fiber reorientation perpendicular to strain. They report that uniaxial stretch rapidly induces autophagy (quantified by LC3 puncta and LC3-I/II conversion) and that inhibition of autophagy (via chloroquine or a CASA-deficient Bag3 mutant) slows actin reorientation. Proteomic analysis after 4 h stretch (plus minus chloroquine) reveals depletion of focal adhesion proteins and CASA regulators, and an enrichment of mitochondrial proteins. Consistent with this, Western blots and immunostaining show decreased levels and size of focal adhesion components (paxillin, vinculin, etc.) under stretch. The authors further report that mitochondrial content (Tomm20 area/number) is unchanged, but staining with MitoTracker Green FM increases after 4 h stretch. They interpret this as increased mitochondrial membrane potential, although cellular ATP levels remain unchanged. Finally, inhibiting ATP synthase with oligomycin

similarly slows stretch-induced actin reorientation. The authors conclude that cyclic stretch triggers CASA-dependent autophagy to remodel focal adhesions and activates mitochondrial function to support the adaptive mechanoreponse.

Novelty and Contribution: The study addresses an important question in mechanobiology: how cells adapt to sustained mechanical stress. Linking cyclic stretch to autophagy-mediated turnover of focal adhesions is novel and supported by a multi-pronged approach (imaging, biochemistry, proteomics). The finding that mitochondria are "activated" by stretch is intriguing and suggests a coordination between energy metabolism and cytoskeletal adaptation. Overall, the work advances our understanding of how mechanotransduction intersects with autophagy and mitochondrial function. The combination of proteomics and functional assays is a strength. However, some conclusions (particularly regarding mitochondrial activation) are not fully substantiated by the data presented.

Methodology and Data Quality: The experimental approaches are generally appropriate. The authors use well-established assays for autophagy (LC3 imaging and flux analysis with chloroquine) and for focal adhesion analysis (immunostaining for paxillin and vinculin). The cyclic stretch protocol (duration, strain amplitude, frequency) could have been more clearly described in the Methods. It is positive that the authors included both immunofluorescence and quantitative proteomics, but some methodological details need clarification:

- **Autophagy Assay:** The quantification of autophagic puncta ("APs") is convincing, and the use of chloroquine to trap LC3-II is appropriate. It would help to state the imaging and analysis criteria for LC3 puncta (e.g. size threshold, normalization to cell area). The decrease in LC3-II without chloroquine after 4 h and its accumulation with chloroquine is consistent with increased flux, but it is not fully demonstrated whether this represents increased autophagy induction or simply turnover. A time course of LC3-II/I with and without lysosomal blockade at earlier times (e.g. 10 min, 1 h) would strengthen the conclusion of early autophagy induction.

• Cytoskeletal Reorientation: The actin fiber orientation analysis seems robust, and the use of confidence intervals is helpful. However, it would clarify interpretation if the authors stated the number of cells analyzed per condition and whether multiple independent experiments were performed.

• Mitochondrial Assays: The authors assess mitochondria by imaging (Tomm20, MitoTracker Green) and by measuring cellular ATP. While Tomm20 area/number shows no change, the MitoTracker staining increased. Importantly, MitoTracker Green FM is generally considered to label mitochondria independently of membrane potential, so its increased intensity is not a definitive indicator of higher potential. The conclusion that membrane potential increased is therefore speculative based on these data. Moreover, ATP levels did not change, raising questions about the functional significance. Direct measurement of membrane potential (e.g. with JC-1) would be a more appropriate assay. Alternatively, an assay of respiration (OCR) or other functional readout could help. The authors do use oligomycin (ATP synthase inhibitor) to show that ATP production contributes to actin reorientation, which supports a metabolic role. However, the link between stretch and mitochondrial "activation" requires more evidence.

Clarity and Figure Presentation: The manuscript is generally well organized and the figures are clear. The labeling in figures (axes, units, scale bars) should be checked carefully. For example, some fluorescence images (e.g. in Figure 1) do not have arrowheads to illustrate stretch direction. In the text, it would be clearer to define term abbreviations at first use (e.g. "FA" for focal adhesion in the abstract). There are a few language issues (e.g. "did not observed differences" should read "did not observe differences" in Results line 269). The narrative generally flows well from autophagy to focal adhesions to mitochondria, although the jump to proteomics and mitochondria is a bit abrupt; perhaps a bridging sentence emphasizing the link to metabolic adaptation would help. Figures 2

and 6 (showing quantitative analyses) are generally well presented. It would be helpful if all quantitative plots included "the number of replicates" in addition to cells analyzed (n) in the legend.

Major Comments:

1. Mitochondrial activity assessment (Figure 7D-E): The reported increase in MitoTracker Green FM staining after 4 h of stretch is intriguing, but the imaging needs careful control. In Figure 7D, the background fluorescence in the stretched vs unstretched images appears different, which could artificially influence the intensity measurement. I strongly recommend showing representative images side-by-side with identical microscope settings (same exposure, gain, and background correction) for the unstretched and stretched conditions. Moreover, include quantification of background-corrected fluorescence intensity (e.g. mean intensity per cell after subtracting background) and indicate the number of cells analyzed. This will ensure that the observed increase is not an artifact of imaging conditions.

Thank you for this comment. First, we want to mention that we have removed the MitoTracker result to the Supplementary material of the manuscript due to the next mentioned major comment (major point 2). We further improved the analysis of the MitoTracker images with improved analysis software. We carefully set a mask to distinguish cell areas from the background and subtracted the mean background signal from the grey values of the masked cells. With that we could quantify background corrected fluorescence intensity in the best possible way as recommended by the reviewer. The representative images are shown in the Figure with subtracted background signals to illustrate the differences. The previously observed difference between unstretched control cells and 4 h stretched cells remained significant, but we decided to move that data to Supplementary Figure S6G and H. The method used to perform the background correction is now described in the material and methods section.

2. Interpretation of MitoTracker signal and suggestion for direct

membrane potential measurement: The authors interpret the increased MitoTracker Green FM signal as evidence of higher mitochondrial membrane potential. However, MitoTracker Green is known to accumulate in mitochondria largely independent of membrane potential; it is often used as a mitochondrial mass marker rather than a potential indicator. The lack of change in mitochondrial number/area (Figure 7A-C) suggests that the signal change could be due to altered dye uptake or mitochondrial membrane composition, not necessarily potential. To robustly test mitochondrial membrane potential, I suggest performing an experiment using a well-established potentiometric dye, such as JC-1. These dyes respond to changes in membrane potential and would directly address whether stretching increases mitochondrial polarization. If new experiments are not feasible before publication, the authors could rephrase conclusions to reflect that MitoTracker intensity changes are correlative and suggest potential changes, rather than definitive proof. At minimum, the authors should acknowledge this limitation more explicitly in the Discussion and moderate the claim that mitochondrial potential is increased.

We agree with the reviewer's interpretation that Mitotracker staining cannot be unambiguously correlated with membrane potential and is more suited to assess mitochondrial mass.

As suggested, we probed the mitochondrial membrane potential in A7r5 ML following stretching using the JC-1 assay. We observed increased mitochondrial membrane potential after 4 h of stretching. We believe these results strengthen the interpretation that mitochondria were activated following stretching. This adds to the increase in mitochondrial protein and transcript levels, and the imaging data presented before. We therefore adjusted Figure 7 to include the JC-1 results, more relevant for the readers, specifically when presented along with the effect of oligomycin inhibition on cytoskeletal orientation.

We moved the Tomm20 and Mitotracker stainings and quantifications to Figure S6D-H.

3. Autophagy flux and temporal dynamics: The data convincingly show a spike in autophagic puncta at 10 min of stretch and normalization by 4 h, with chloroquine causing sustained puncta. While the LC3-I/II blots support increased turnover, the interpretation would be stronger with a clearer depiction of flux. For example, showing LC3 blots (± chloroquine) at the 10-min time point could confirm that the early autophagy spike is real. The current data focus on 4 h with/without CQ. The authors should also clarify how "APs" were counted in imaging (e.g. criteria for a punctum) and note whether any differences in cell morphology under CQ could affect puncta counts. Overall, explicitly stating that stretch induces a transient burst of autophagy (peaking at ~10 min) would help clarify the dynamic nature of this response.

We highly appreciate the reviewer's suggestion to provide further evidence regarding the stretching-induced autophagic flux by additional experiments.

As suggested, we have analyzed blots of LC3B-I and -II after 10 min of stretching without and with CQ treatment (Figure S2A-B). Here, we could observe a slight increase in LC3B-II after 10 min of stretching which was even higher with CQ treatment. Most likely due to low number of repeats we could not detect significant differences.

Therefore, we further analyzed the LC3B-I and II levels across the stretching timeframe (10, 30, 60, 240 min). We used CQ treatment in both unstr. and 4 h str. conditions as a control for impaired flux. Stretching lead to a quick increase of LC3B-II relative to LC3B-I, which returned after 4 h of stretching (Figure 2C-D. CQ treatment led to accumulation of the lipidated (membrane form) of LC3B, consistent with the impairment of their turnover observed by imaging. As before, we consistently observed that the levels of both LC3B-II and LC3B-I were lower after 4 h of stretching. Taken together, these results add to the microscopy and the proteomics data, arguing for a functional engagement of autophagy upon stretching, where the autophagic flux, measured by autophagosome formation (higher lc3b-II/lc3b-I ratio) and consumption (lower ratio and lower levels of both lc3b-II and lc3b-I)

are noticeable. The new data has been added to Figure 2 and we have stated that stretch induces a transient burst of autophagy as suggested by the reviewer.

Regarding the AP counting:

As we have analyzed the number of LC3B-spots per cell in monolayer cells, we were not able to quantify the cell area for normalization. Therefore, we have counted the cell nuclei per image and normalized the number of LC3B-spots to the nuclei number per image rather than normalizing it to the whole image area, which would have been less accurate in our eyes. We have clarified the analysis more precise in the Methods section 'LC3B-spots analysis' to avoid misunderstandings.

With the normalization to cell nuclei, we also expect to avoid affecting the LC3B-spots per cell area due to effects on the cell morphology e.g. shrinkage of cell area due to CQ treatment.

4. Mechanistic clarity and CASA involvement: The manuscript emphasizes the role of Bag3-mediated CASA in this process. The use of a phospho-mimetic Bag3 mutant is a strength. However, it would help to clearly delineate which observations are specific to CASA. For instance, the proteomics shows depletion of CASA regulators (Bag3, Hsp8, etc.) after stretch. Are any of the accumulated mitochondrial proteins known Bag3 clients, or is there a hypothesized link between CASA and mitochondrial regulation? The authors might consider a brief discussion of whether CASA (or chaperone-mediated pathways) could directly influence mitochondrial quality control or biogenesis under mechanical stress. As is, the link is more correlative. Emphasizing the distinction between CASA's effect on cytoskeletal proteins and the separate effect on mitochondria would improve mechanistic clarity.

We greatly appreciate this interesting comment regarding the roles of Bag3-mediated autophagy (CASA) in the cellular responses to cyclic stretching, including cytoskeletal reorientation and

mitochondrial activation).

Although a direct link between CASA and mitochondrial protein degradation was not yet established, we previously observed that a dephosphorylation-mimicking mutant form of Bag3 (CASA-active form) interacts with several mitochondrial proteins (Figure 3E, Ottensmeyer et al. 2024). This aligns with the altered mitochondrial protein levels observed in the hearts of a dysfunctional Bag3 (P209L) mutant expressing mice (Kimura et al., 2021). In addition, autophagosome formation during CASA depends on the interaction between Bag3 and Rab7a, a Rab GTPase implicated in the early steps of mitophagy (Heo et al., 2018; Yan et al., 2022). Strikingly, we also found that Bag3 dephosphorylation (activation) is triggered by chemical induction of mitophagy (Figure S5B and S5C, Ottensmeyer et al. 2024), consistent with previous observations of Bag3 recruitment to depolarized mitochondria (Tahrir et al., 2017).

We believe that Bag3 dephosphorylation (Figure 1E, Ottensmeyer et al. 2024) regulates its function under stretching (Figure 3, this study). It is conceivable that spatial regulation of Bag3, targeting focal adhesions (and other mechanosensory structures) under strain, reduces its availability to participate in mitophagy. We did not emphasize clearly enough that the accumulation of mitochondrial proteins (and biomass) observed during stretching does not simply result from reduced mitophagy, but that the activation of mitochondria actively contributes to the adaptation. Considering the added data on membrane potential, it is now clearer that mitochondria activity is induced, and involved in the adaptation (Figure 7, this study).

Accordingly, we revised the final paragraph of the discussion to incorporate these points and references, highlighting how chaperone-mediated pathways may influence mitochondrial quality control, biogenesis, and ultimately their activity under strain.

5. Introduction - discuss cytoskeletal control of ATP. The Introduction frames a link between cytoskeleton and metabolism, but it would benefit from citing very recent work on this topic.

Several studies have established that mechanical or cytoskeletal cues directly regulate glycolytic flux and ATP production. For example, Hu et al. showed that PI3K-mediated actin remodeling releases aldolase A from F-actin, leading to a boost in glycolysis (PMID: 26824656). Similarly, Park et al. demonstrated that substrate stiffness mechanically controls glycolysis: softening the matrix caused stress fiber disassembly and degradation of phosphofructokinase, thereby downregulating glycolysis (PMID: 32051585). More recently, Zhan et al. (PMID: 38328193) described dynamic cortical "glycolytic waves" driven by Ras/PI3K-F-actin networks, which locally supply ATP for protrusion and migration. These findings underscore that cytoskeletal architecture and tension influence ATP levels and metabolic enzymes. The authors are encouraged to briefly discuss these studies in the Introduction (or Discussion) to place their mitochondria and ATP results in context.

We are very grateful for pointing us to these highly relevant references. We now compare our results to this novel and sophisticated data about the interconnections between cytoskeletal architecture, tension, protrusions and metabolic activity through different pathways. We agree that the literature mentioned increases the value of the data and the discussion presented here.

Minor Comments:

- In the Abbreviations section, "AP, autophagosome" and "ATP, adenosine triphosphate" appear on the same line, which is confusing. Please separate them clearly. Also define "APs" as autophagosomes at first use in the main text.

Thank you. We corrected this mistake and define APs as autophagosomes at the first time used in the main text in section 'Uniaxial stretch induces autophagy'.

- On p.2 (lines 267-269), correct the grammar: "we did not observed differences" should be "we did not observe differences".

Thank you. Corrected as suggested.

- In the Methods, please specify the parameters of the cyclic stretch (amplitude, frequency, duty cycle). This is important for reproducibility.

The parameters of cyclic stretching including amplitude and frequency are added to the Methods section part 'Cell Stretching' to ensure reproducibility of the experiments.

- In Figure legends, always state the number of biological replicates (n) or cells analyzed. For example, in Figure 2 and 6, indicate how many cells or wells contributed to the quantified data.

The number of biological independent replicates have been added to all figure legends and supplementary figure legends. Moreover, we have indicated how many cells contributed to the quantified data in Figure 2 and 6.

- In Figure 1 and other orientation plots, clarify how the angle distribution was computed (e.g. direction of stretch = 0{degree sign}, parallel = 0-5{degree sign} = "parallel", etc.). Also, mention the total number of cells analyzed per condition.

We have clarified the direction of stretch in correlation to the orientation in degree in the cumulative frequency plot in all Figures for better understanding: 'Actin fiber orientation is represented as an angular distribution from 0° to 90° in the direction of stretch (**0° means in stretch direction; 90° means perpendicular to stretch direction**), plotted as cumulative frequencies.'

The number of analyzed cells is given as N in the figure legends for the reorientation analysis of single cells. For monolayer analysis the number of images analyzed with typically >15 cells per image is provided.

- The discussion could briefly mention other studies linking

mechanotransduction and mitochondrial function (e.g. via calcium signaling or ER stress) to contextualize the findings.

We have addressed that point by adding current literature about mechanically induced calcium signaling and ER stress setting our results into these contexts and discussing potential mechanical concordance explaining our findings.

- Ensure statistical tests and p-values are described in the Methods figure legends. Some figures label "*" for $p < 0.05$; please define these and mention the test used in the legends.

We have added information about the statistical test used as well as p-values to all figure legends and supplementary figure legends as recommended.

Recommendation: The manuscript presents an interesting study at the intersection of mechanobiology, autophagy, and mitochondrial biology. The experimental work is largely sound and the multi-pronged approach is comprehensive. However, the interpretation of mitochondrial activation needs strengthening, and some quantitative clarifications (as noted above) are required. I recommend major revision. Addressing the key points about mitochondrial assays (adding or revising data on membrane potential, ensuring imaging controls for Figure 7) and clarifying the autophagy-focal adhesion link will greatly improve the rigor and clarity of the study. Upon revision, the manuscript will make a valuable contribution to understanding how cells adapt to mechanical stretch.

Thank you very much for the summary of recommendations and the statement that the revised manuscript will make a valuable contribution for the understanding of cellular adaptation to mechanical stress.

December 22, 2025

RE: Life Science Alliance Manuscript #LSA-2025-03347-TR

Dr. Bernd Hoffmann
Forschungszentrum Jülich
Forschungszentrum Jülich GmbH/ICS-7 Biomechanics
Research Centre Juelich, Institute of Bio- and Nanosystems, IBN-4
Jülich 52425
Germany

Dear Dr. Hoffmann,

Thank you for submitting your revised manuscript entitled "Cyclic stretch induces autophagy-mediated focal adhesion remodeling and activates mitochondria". As you will see, reviewers are overall satisfied. We appreciate the concern of Reviewer 1 on protein loading and we invite you to consider any text changes regarding protein normalization, if needed. In our routine screening to which all manuscripts are subjected, we found a few images in your manuscript that appear very similar as some images from a prior paper from your group ("Strain-induced mechanoresponse depends on cell contractility and BAG3-mediated autophagy"). These are Figure 1C (top left image) and Figure 4A (two images in the top row). Please clarify if the data shown in this work are the result of original observations, and if so please select different example images. Pending resolution of this question we would be happy to publish your paper in Life Science Alliance after final revisions necessary to meet our formatting guidelines.

- Please be sure that the authorship listing and order is correct.
- Please add ORCID ID for secondary corresponding author - they should have received instructions on how to do so.
- Please add the X and Bluesky handles of your host institute/organization, as well as your own and/or one of the authors, in our system.
- Please upload a clean manuscript file, without the track changes. The version that highlights the changes you may upload with the file designation "Related Manuscript File".
- The image shown in Figure 6H (second row, middle image) appears to depict the same cell as shown in Figure 6A (last row, middle image). Please acknowledge the reuse of this image in the figure legend.

A. FINAL FILES:

B. MANUSCRIPT ORGANIZATION AND FORMATTING:

Thank you for your attention to these final processing requirements. Please revise and format the manuscript and upload materials as soon as you are able.

Sincerely,

Reviewer #1 (Comments to the Authors (Required)):

The revised manuscript now includes autophagic flux, imaging data on mitochondrial potential and a revised conclusion section to which extent focal adhesion proteins are affected by cyclic stretch induced autophagy. These points clarify my main concerns.

As technical point pending, it is unclear whether the data in Fig. S4B and S4D were normalized to an appropriate loading control, i.e. a protein not affected by autophagy? Because interference with autophagy may alter the overall protein abundance in the cell, it is questionable whether normalizing to Ponceau S intensity is appropriate.

Reviewer #2 (Comments to the Authors (Required)):

The authors have done an excellent job to rigorously address my comments. I am satisfied that they have answered all of my questions and I have no further comments. I would like to congratulate the authors on a thorough and interesting study.

January 19, 2026

RE: Life Science Alliance Manuscript #LSA-2025-03347-TRR

Dr. Bernd Hoffmann
Forschungszentrum Jülich
Forschungszentrum Jülich GmbH/ICS-7 Biomechanics
Research Centre Juelich, Institute of Bio- and Nanosystems, IBN-4
Jülich 52425
Germany

Dear Dr. Hoffmann,

Thank you for submitting your Research Article entitled "Cyclic stretch induces autophagy-mediated focal adhesion remodeling and activates mitochondria". It is a pleasure to let you know that your manuscript is now accepted for publication in Life Science Alliance. Congratulations on this interesting work.

Your manuscript will now progress through copyediting and proofing. Please refer to our earlier email inviting a slight change to the text at line 158.

DISTRIBUTION OF MATERIALS:

Again, congratulations on a very nice paper. I hope you found the review process to be constructive and are pleased with how the manuscript was handled editorially. We look forward to future exciting submissions from your lab.

Sincerely,
